# In-plane strain engineering in ultrathin noble metal nanosheets boosts the intrinsic electrocatalytic hydrogen evolution activity

Geng Wu[1,5], Xiao Han[1,5], Jinyan Cai[1,5], Peiqun Yin[1], Peixin Cui [2], Xusheng Zheng[3], Hai Li [4], Cai Chen[1], Gongming Wang [1✉] & Xun Hong [1✉]

Strain has been shown to modulate the electronic structure of noble metal nanomaterials and alter their catalytic performances. Since strain is spatially dependent, it is challenging to expose the active strained interfaces by structural engineering with atomic precision. Herein, we report a facile method to manipulate the planar strain in ultrathin noble metal nanosheets by constructing amorphous–crystalline phase boundaries that can expose the active strained interfaces. Geometric-phase analysis and electron diffraction profile demonstrate the in-plane amorphous–crystalline boundaries can induce about 4% surface tensile strain in the nanosheets. The strained Ir nanosheets display substantially enhanced intrinsic activity toward the hydrogen evolution reaction electrocatalysis with a turnover frequency value 4.5-fold higher than the benchmark Pt/C catalyst. Density functional theory calculations verify that the tensile strain optimizes the $d$-band states and hydrogen adsorption properties of the strained Ir nanosheets to improve catalysis. Furthermore, the in-plane strain engineering method is demonstrated to be a general approach to boost the hydrogen evolution performance of Ru and Rh nanosheets.

[1] Center of Advanced Nanocatalysis (CAN), Department of Applied Chemistry, Hefei National Laboratory for Physical Sciences at the Microscale, University of Science and Technology of China, Hefei, Anhui 230026, P.R. China. [2] Key Laboratory of Soil Environment and Pollution Remediation, Institute of Soil Science, Chinese Academy of Sciences, Nanjing, Jiangsu 210008, P.R. China. [3] National Synchrotron Radiation Laboratory (NSRL), University of Science and Technology of China, Hefei, Anhui 230029, P.R. China. [4] Key Laboratory of Flexible Electronics (KLOFE) & Institute of Advanced Materials (IAM), Jiangsu National Synergetic Innovation Center for Advanced Materials (SICAM), Nanjing Technology University, Nanjing, Jiangsu 211816, P.R. China. [5] These authors contributed equally: Geng Wu, Xiao Han, Jinyan Cai. ✉email: wanggm@ustc.edu.cn; hongxun@ustc.edu.cn

Noble metal nanomaterials have shown significant application towards catalysis, electronics and photonics[1–4], which have attracted extensive attention. To boost the utilization efficiency of noble metal, tremendous investigations have demonstrated that tuning atomic-level structures of noble metal nanomaterials can enhance their catalytic performance[5–10]. Among them, ultrathin nanosheets possess intrinsic merits including high specific surface area and abundant exposure of surface active atoms with low-coordination-number[11–15], which make them exceedingly desirable for catalysis.

Moreover, since surface strain can modulate the electronic structure of catalysts by shifting the $d$-band centre, which is closely related with the adsorption energy and activation energy barriers of reaction intermediates during catalytic process[16–19], constructing surface strain in ultrathin noble metal nanosheets can be the promising avenue to enhance their intrinsic activity. Meanwhile, strain could also create high-energy active interfaces distinct from equilibrium thermodynamic state[20–22], which have been predicted to be highly active for catalysis[22]. Generally, surface strain is generated through lattice mismatch which extensively exists in core-shell nanostructure or metal-substrate interface[23–28](Fig. 1a, b). For example, the intermetallic PtPb core can induce conformal four-layer Pt shell with tensile strain of 7.5% in PtPb/Pt core-shell nanoplates, which enable the catalyst with optimal oxygen adsorption energies and thus superior performance of oxygen reduction reaction[23]. Besides, the strain in ultrasmall Pt nanoparticles can be tuned from compressive strain to tensile strain by altering lattice spacing of $LiCoO_2$ support and further optimize their catalytic performance[26]. However, the strain induced by lattice mismatch is spatially dependent, which generally decays from the interface toward the outmost surface and vanishes beyond several atomic layers[29]. Therefore, the catalytic activity is highly related to the thickness of shell in core-shell structure or the size of metal in metal-substrate structure. Given the fact that the strained interface is typically embedded in the core-shell nanostructures or metal-substrate structure, the created active interface is basically unexposed for catalytic reaction. In this regard, developing in-plane strain engineering in ultrathin nanosheet could fully expose the active regions for catalysis (Fig. 1c). To this end, lateral structural engineering with atomic level precision is the key to achieving planar strain distribution in ultrathin nanosheet, but remains a great challenge.

Herein, we report a facile method to engineer the planar strain in ultrathin noble metal nanosheets by constructing high-density in-plane amorphous–crystalline phase boundaries. Taking ultrathin amorphous–crystalline Ir nanosheets (denoted

as AC-Ir NSs) with thickness of 5 nm as example, the in-plane amorphous–crystalline boundaries can enable the nanosheets to generate about 4% tensile strain. Furthermore, the tensile strain can efficiently modify the $d$-band state of Ir sites and thus enhance HER performance of AC-Ir NSs. The capability to manipulate the planar strain engineering with controlled spatial distribution could pave new avenues toward catalytic application and beyond.

## Results

**Synthesis and characterization of AC-Ir NSs.** The AC-Ir NSs have been synthesized at annealing temperature of 300 ºC via a salt-assisted method. The crystallinity of Ir nanosheets could be tuned by controlling annealing temperature (Supplementary Fig. 1). Nanosheets with a lateral size of several micrometers were confirmed by transmission electron microscopy (TEM) image, scanning electron microscopy (SEM) images and high-angle annular dark-field scanning transmission electron microscopy (HAADF-STEM) image (Fig. 2a and Supplementary Fig. 2 and 3). As revealed in atomic force microscopy (AFM) image (Fig. 2b), the thickness of nanosheet was measured to be about 5 nm, verifying their ultrathin feature. Furthermore, the scanning transmission electron microscopy energy-dispersive X-ray spectroscopy (STEM-EDS) elemental mappings (Fig. 2c) display the homogeneous distribution of Ir and C elements over the nanosheets. By contrast, a tiny amount of oxygen element was also detected on the nanosheets because of the unavoidable air exposure. According to the compositional analysis by quantitative EDS spectrum (Supplementary Fig. 4), the Ir content in nanosheets was determined to be approximately 80 wt%. To further investigate the atomic structure of the obtained Ir nanosheets, high-resolution transmission electron microscopy (HRTEM) images (Fig. 2d and Supplementary Fig. 5) demonstrated the nanosheets are composed of crystalline and amorphous domains. The diffuse-ring in the selected-area fast Fourier transform (FFT) pattern (Fig. 2e$_1$) and the corresponding inverse fast Fourier transformation (IFFT) image (Fig. 2e$_2$) of amorphous domain confirmed the disordered distribution of Ir atoms. The bright spots in the FFT pattern (Fig. 2f$_1$) from crystalline domain belonged to face-centered-cubic ($fcc$) [0−11] diffractogram. As displayed in IFFT image of crystalline domain (Fig. 2f$_2$), the lattice spacing of Ir (111) plane is 0.23 nm, which is slight larger than that of Ir crystal. Furthermore, the amorphous–crystalline boundary has been verified by FFT pattern (Fig. 2g$_1$) with the coexistence of diffuse-ring and diffraction spot, IFFT image (Fig. 2g$_2$) and HRTEM images (Supplementary Fig. 6). To further probe the element distribution of amorphous and crystalline domain in AC-Ir NSs, the high-resolution EELS spectroscopy has been performed. As shown in Fig. 2i and Supplementary Fig. 7, the signal of C is mainly observed in the amorphous domains, while C signal is almost absent in the crystalline domains. The O signal is very limited in both amorphous domains and crystalline domains, revealing amorphous domain of AC-Ir NSs could be C-containing amorphous iridium structure.

**Strain analysis around amorphous–crystalline boundary.** As shown in Fig. 3a, an amorphous–crystalline phase boundary can be observed, where a structural break across the interface is marked by the white dashed line. To explore the distribution and extent of lattice strain across the crystalline domain in AC-Ir NSs, geometric phase analysis (GPA) was conducted[30,31]. From Figs. 3b, c, the strain values ($e_{yy}$) gradually reduce as the distance away from phase boundary and the tensile strain in crystalline domain mainly concentrates around 4%. Moreover, to explore the lattice strain across whole nanosheets, the electron diffraction

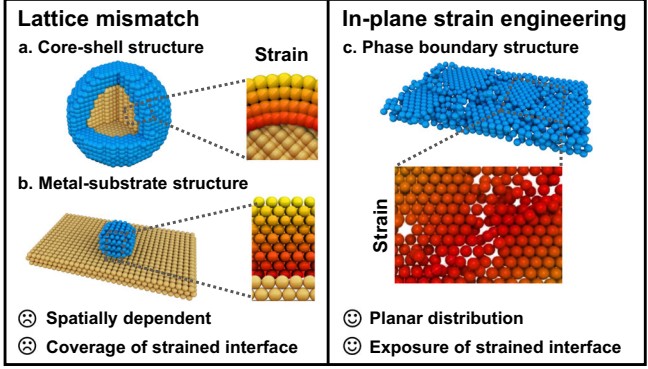

**Fig. 1 Schematic illustration of strain engineering. a**, **b** Strain engineering through the lattice mismatch approach in core-shell structure or metal-substrate structure, respectively. **c** The in-plane strain engineering through constructing amorphous–crystalline phase boundaries.

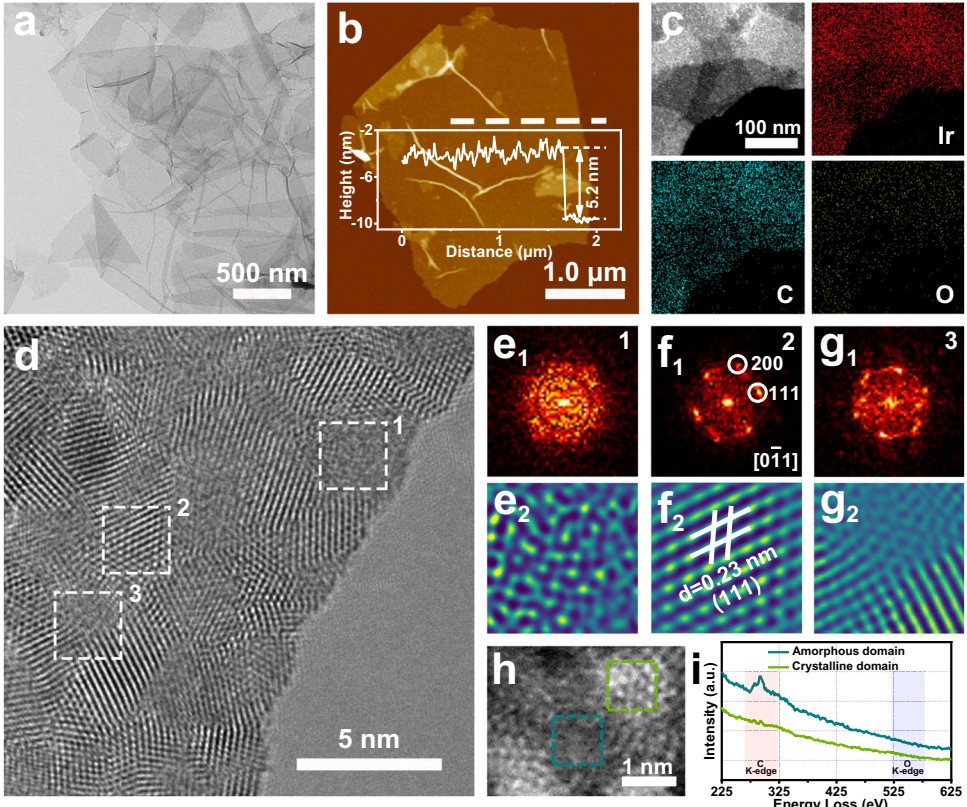

**Fig. 2 Characterizations of AC-Ir NSs. a** TEM image, (**b**) AFM image, (**c**) EDS elemental mapping images and (**d**) HRTEM image of AC-Ir NSs. **e₁, f₁, g₁** FFT patterns, (**e₂, f₂, g₂**) IFFT images of amorphous domain, crystalline domain and amorphous–crystalline phase boundary, respectively. **h** Aberration-corrected HAADF-STEM image around amorphous–crystalline phase boundary (inset: dark cyan: amorphous domain, dark green: crystalline domain.). **i** EELS spectroscopy of amorphous domain and crystalline domain corresponding to the selected area in (**h**), respectively. Note: AC-Ir NSs were obtained at annealing temperature of 300 °C.

profile (EDP) was further employed through rotationally averaging integration of electron diffraction pattern (Fig. 3d–f)[32,33]. As shown in Fig. 3d, the diffraction peaks of AC-Ir NSs show notable negative shifts compared with Ir crystals and the lattice expansion of AC-Ir NSs is estimated to be around 4%. Moreover, from X-ray diffraction (XRD) pattern (Supplementary Fig. 9), three diffraction peaks of AC-Ir NSs shows negative shifts relative to the Ir crystal, also confirming the tensile strain can be generated in AC-Ir NSs. Combining the analysis of GPA, EPD and XRD, it is convincing to conclude amorphous–crystalline boundary can introduce tensile strain in the nanosheet.

**Electronic structure and coordination information analysis of AC-Ir NSs.** In order to elucidate the local electronic and coordination structural information of AC-Ir NSs, X-ray absorption fine structure (XAFS) measurements were employed. Compared with Ir powder and commercial $IrO_2$, the X-ray absorption near-edge structure (XANES) spectrum and the first derivative XANES spectrum (Fig. 4a and Supplementary Fig. 10) exhibit the average electronic state of the Ir in AC-Ir NSs is situated between the metallic state (Ir powder) and oxidative state ($IrO_2$)[34], indicating the partially oxidation of the AC-Ir NSs. As shown in the Fourier transformed extended X-ray absorption fine structure (EXAFS) spectrum (Fig. 4b and Supplementary Table 1), the peak at about 2.9 Å is ascribed to Ir-Ir bond and the peak appearing at about 2.0 Å is associated with the Ir-C bond, which mainly originates from C-containing amorphous Ir domains. To further distinguish the bonding between the iridium and the light elements, the wavelet transform spectra of AC-Ir NSs was performed[35]. As

shown in Fig. 4c, compared with the intensity at about 7 Å$^{-1}$ from Ir-O coordination of commercial $IrO_2$, the intensity at around 6 Å$^{-1}$ is attributed to Ir-C coordination, which accords with the calculated Ir-C path ($k = 6.1$ Å$^{-1}$) instead of the calculated Ir-O path ($k = 6.8$ Å$^{-1}$) (Supplementary Fig. 11). Besides, the intensity around 11 Å$^{-1}$ is corresponded to Ir-Ir coordination, which resembled the Ir-Ir coordination in Ir powder, suggesting the existence of iridium oxide and iridium carbide in the AC-Ir NSs are very limited. Furthermore, from the Ir L₃-edge $k^3\chi(k)$ oscillation curves (Supplementary Fig. 12), the oscillation feature of AC-Ir NSs is very similar to that of Ir powder and apparently different from that of $IrO_2$, indicating the similar atomic configuration for Ir site in both AC-Ir NSs and metallic Ir, which coincides with the wavelet transform spectra.

In addition, as depicted in Ir 4f X-ray photoelectron spectroscopy (XPS) spectrum (Fig. 4d), the metallic state and oxidative state of Ir species are concomitant in AC-Ir NS, in which the metallic Ir occupies the dominant species. Furthermore, the oxidized peaks of Ir in AC-Ir NSs are located at 62.2 and 65.0 eV, which are negatively shifted in comparison with commercial $IrO_2$ (62.5 eV and 65.4 eV), suggesting the chemical states of Ir in AC-Ir NSs are different from the Ir-O bond in $IrO_2$. To explore the electronic structure influenced by planar strain, synchrotron-based valence band spectrum was performed, in which the peaks located between 0 and 2 eV in valence band spectrum could be ascribed to the metal d-band[36,37]. For comparison, amorphous Ir NSs (A-Ir NSs, Supplementary Fig. 13) and crystalline Ir NSs (C-Ir NSs, Supplementary Fig. 14) were also conducted. From Fig. 4e, the valence band maximum values are determined to be

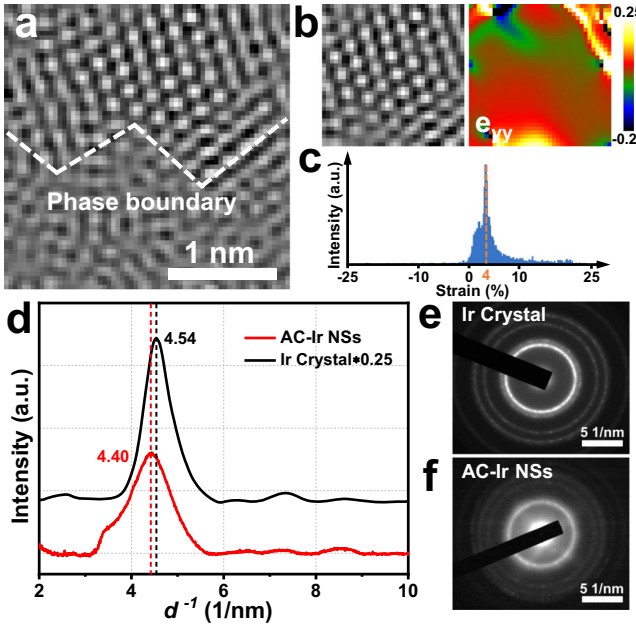

**Fig. 3 Strain distribution across the amorphous–crystalline boundary in AC-Ir NSs. a** HRTEM image of amorphous–crystalline boundary. **b** HRTEM image (left) and $e_{yy}$ strain component (right) of crystalline domain. **c** The histogram of strain distribution. **d** Electron diffraction profile of AC-Ir NSs and Ir crystal. **e, f** Selected area electron diffraction pattern (SAED) of Ir crystal and AC-Ir NSs, respectively. Note: the lattice strain $e_{yy}$ is associated with the expansion/contraction of the lattice vector along the in-plane y direction.

approximately 0.68 eV, 0.52 eV and 1.45 eV for AC-Ir NSs, C-Ir NSs and A-Ir NSs, respectively. The metal $d$-band maximum respecting to the Fermi level ($E_F$) in AC-Ir NSs show positive shifts relative to the C-Ir NSs, which illuminates the planar strain can modulate the $d$-band state of Ir sites in AC-Ir NSs.

**Understanding the effect of tensile strain.** Density functional theory (DFT) calculation of AC-Ir NSs, C-Ir NSs and A-Ir NSs were performed to uncover the effects of the strained electronic structures on surface adsorption behavior. Based on the structural information obtained by HRTEM, GPA and XAFS characterization, the atomic models of AC-Ir NSs, C-Ir NSs and A-Ir NSs was constructed (Supplementary Fig. 15, 16 and Theoretical calculation methods). As displayed in the surficial deformation electron density analysis (Fig. 5a and Supplementary Fig. 17), the electron density of amorphous domains is obviously stronger than that of crystalline domains, indicating the electrons transfer from crystalline domains to amorphous domains in AC-Ir NSs. This result is further confirmed by the Bader charge analysis (Supplementary Table 2), in which the calculated Bader charge transfer from crystalline domains to amorphous domains are 1.52, 0.7, 0.7, 0.67, 0.65, 1.39 eV for AC-Ir NSs with tensile strain from 0% to 20%, respectively. The partial density of states (PDOS) of Ir sites in AC-Ir NSs were calculated to illustrate the modulation of electronic $d$-band featured by planar tensile strain. Figure 5b reveals the $d_{xz+yz+z}^2$ band center of Ir on AC-Ir surface with 4% tensile strain ($-2.16$ eV) is located between the values of C-Ir NSs ($-1.95$ eV) and A-Ir NSs ($-2.42$ eV), which coincides well with the synchrotron-based valence band spectrum results. The PDOS result implies the anti-bonding state downshifts and more electron filling of metal-H anti-bonding orbital in AC-Ir NSs as relative to C-Ir NSs, which afford that AC-Ir NSs probably

possess relatively weak Ir-H interaction and superior electro-catalytic HER performance. To support our conclusions, the standard hydrogen adsorption Gibbs free energy ($\Delta G_{H*}$), which is considered as the descriptor for HER activity, on the surface of AC-Ir NSs was also performed (Fig. 5c and Supplementary Fig. 18, 19). Figure 5c reveals the H$^*$ adsorption on surface of AC-Ir NSs with 4% tensile strain ($-0.06$ eV) is more optimal than those on A-Ir NSs (0.22 eV), C-Ir NSs ($-0.37$ eV) and Pt/C catalysts ($-0.23$ eV), based on the conclusions of the HER volcano plot[38,39], which suggests AC-Ir NSs with 4% tensile strain possess highly intrinsic HER activity. From Supplementary Fig. 20–22, the strained AC-Ir NSs possess more Ir active sites with favorable $\Delta G_{H*}$ compared with AC-Ir NSs without tensile strain, suggesting that the tensile strain can effectively improve the number of active sites in AC-Ir NSs. Additionally, the $d_{xz+yz+z}^2$ band center of the Ir sites and the $\Delta G_{H*}$ of AC-Ir NSs as a function of tensile strain are summarized in Fig. 5d and Supplementary Fig. 23, 24. It shows a rough trend in upward shift of the band centers as the tensile strain increases in AC-Ir NSs, which always downshifts compared to C-Ir NSs. Meanwhile, the $\Delta G_{H*}$ reaches the optimal value at around 4% tensile strain among different degrees of lattice expansion, revealing the 4% tensile strain plays an important role on enhancing the intrinsic HER activity. Moreover, considering that the AC-Ir NSs contains a small amount of carbon (C) element, the DFT calculations of C-doped crystalline Ir NSs were performed to further explore the influence of the C dopants on the intrinsic HER activity (Supplementary Fig. 25). The C-doped crystalline Ir NSs possess more negative $\Delta G_{H*}$ ($-0.29$ eV) than AC-Ir NSs with 4% tensile strain ($-0.06$ eV), suggesting the improved intrinsic activity is not mainly stemmed from carbon dopants. These DFT results suggest the intrinsic HER activity of AC-Ir NSs could be mainly stemmed from 4% tensile strain induced by amorphous–crystalline boundary instead of the carbon dopants.

**Electrocatalytic activity of AC-Ir NSs towards HER.** To evaluate the effects of strain on the catalytic properties, the HER of AC-Ir NSs in acidic media was conducted. For comparison, A-Ir NSs, C-Ir NSs and commercial Pt/C catalysts (Supplementary Fig. 26) were also assessed. As shown in polarization curves (Fig. 6a), the AC-Ir NSs require much lower overpotential (17 mV versus reversible hydrogen electrode (*vs*. RHE)) to achieve current density of 10 mA cm$^{-2}$ than that of A-Ir NSs (65 mV), C-Ir NSs (32 mV) and Pt/C catalysts (20 mV). To illuminate the underlying reaction kinetics, the corresponding Tafel plots were achieved (Fig. 6b). The resulting Tafel slope of AC-Ir NSs (27 mV dec$^{-1}$) is slightly lower than that of Pt/C catalysts (29 mV dec$^{-1}$), which demonstrates that the HER pathways of AC-Ir NSs follow the Volmer-Tafel mechanism with dissociation for adsorbed hydrogen atoms as the rate-determining step[40]. Moreover, the exchange current density of AC-Ir NSs (1.78 mA cm$^{-2}$) is considerably larger than that of A-Ir NSs (0.32 mA cm$^{-2}$), C-Ir NSs (1.03 mA cm$^{-2}$) and Pt/C catalysts (1.36 mA cm$^{-2}$) (Fig. 6c), indicating AC-Ir NSs possess faster kinetics and higher intrinsic activity towards HER among all the tested catalysts. Remarkably, AC-Ir NSs achieve a mass activity of 1503.2 A g$^{-1}$ at overpotential of 50 mV (*vs*. RHE), which is 2.2-fold, 6.7-fold and 47.0-fold as high as Pt/C catalysts (694.4 A g$^{-1}$), C-Ir NSs (222.9 A g$^{-1}$) and A-Ir NSs (32.0 A g$^{-1}$), respectively (Fig. 6d and Supplementary Fig. 27). Importantly, AC-Ir NSs deliver the smallest overpotential of 17 mV at the current density of 10 mA cm$^{-2}$ among the recently reported noble metal-based HER electrocatalysts[40–49] (Fig. 6e and Supplementary Table 3).

To further elucidate the outstanding HER performance, the electrochemical impedance spectroscopy (EIS) was measured. From

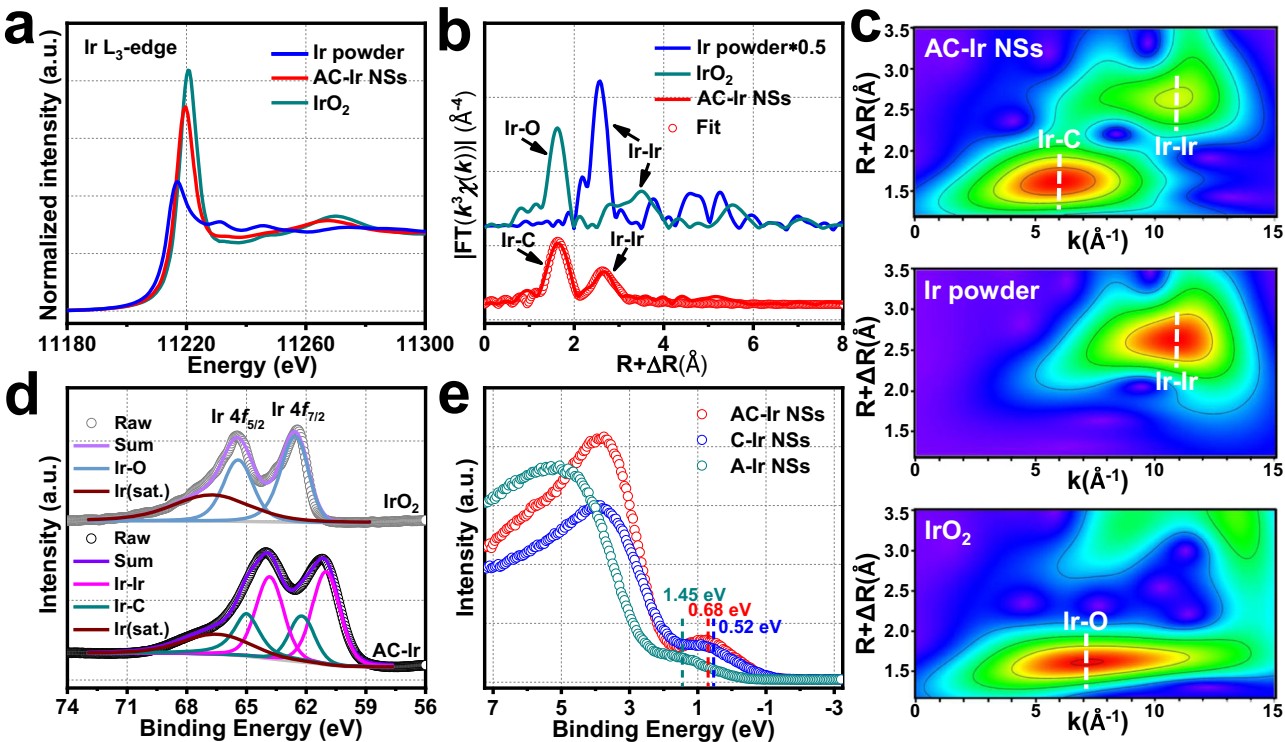

**Fig. 4 Electronic structure and coordination information of AC-Ir NSs. a** XANES spectrum, (**b**) Fourier transformed of k³-weighted χ(k)-function of the EXAFS spectrum and (**c**) The wavelet transform spectra of AC-Ir NSs, Ir powder and commercial IrO₂, respectively. **d** Ir 4 f XPS spectrum of AC-Ir NSs and commercial IrO₂. **e** The synchrotron-based valence band spectrum of AC-Ir NSs, C-Ir NSs and A-Ir NSs, respectively.

Supplementary Fig. 28, AC-Ir NSs have lower charge-transfer resistance than A-Ir NSs and C-Ir NSs, revealing AC-Ir NSs possess rapid charge transfer kinetic. Moreover, as displayed in Fig. 6f and Supplementary Table 4, AC-Ir NSs deliver a high turnover frequency (TOF) value of 3.61 $H_2$ $s^{-1}$ at an overpotential of 30 mV (vs. RHE), which is 4.5-fold and 6.0-fold than Pt/C catalysts (0.81 $H_2$ $s^{-1}$) and C-Ir NSs (0.60 $H_2$ $s^{-1}$). Combining the data of Tafel slope, mass activity and TOF, it is convincing to validate the in-plane strain engineering have substantially improved the intrinsic activity of the strained Ir nanosheets. Besides activity, the durability of AC-Ir NSs was assessed by accelerated durability test (ADT) and chronoamperometry measurement. As depicted in Fig. 6g, the activity of AC-Ir NSs exhibits negligible degradation after 20,000 cycles, whereas the HER activity of Pt/C catalysts displays significant deterioration after 20,000 cycles. Furthermore, the chronoamperometry measurement (Fig. 6h) also confirm AC-Ir NSs possess superior electrochemical stability, in which the current density of AC-Ir NSs shows ignorable degradation after 18-hour test. Impressively, the electrochemical active surface area, structure and tensile strain of AC-Ir NSs after ADT remain unchanged, underscoring the structural robustness (Supplementary Fig. 31–34). Furthermore, the alkaline HER perform of AC-Ir NSs was also investigated (Supplementary Fig. 35). AC-Ir NSs require an overpotential of 27 mV to afford a current density of 10 mA $cm^{-2}$, which was lower than that of A-Ir NSs (148 mV), C-Ir NSs (188 mV) and Pt/C catalysts (34 mV), suggesting in-plane strain engineering can also boost the alkaline HER catalysis of Ir nanosheets.

**HER catalysis of other amorphous–crystalline noble metal nanosheets.** To evaluate the generality of the in-plane strain induced by amorphous–crystalline boundary for improving the HER catalysis of other noble metal nanosheets, amorphous–crystalline Ru

nanosheets (AC-Ru NSs) and amorphous–crystalline Rh nanosheets (AC-Rh NSs) were prepared and their HER performance also were studied (Supplementary Fig. 36–41). As shown in Supplementary Fig. 38, the AC-Ru NSs require an overpotential of 21 mV at the current of 10 mA $cm^{-2}$ in alkaline medium, which was much less than that of A-Ru NSs (38 mV), C-Ru NSs (38 mV) and Pt/C catalysts (34 mV). Similarly, the AC-Rh NSs exhibit superior electrocatalytic HER performance, which outperforms A-Rh NSs, C-Rh NSs, commercial Pt/C catalysts and most of the reported state-of-the-art noble metal catalysts (Supplementary Fig. 41 and Supplementary Table 5). These results suggest that the in-plane strain induced by amorphous–crystalline phase boundaries is a general strategy to boost the HER catalysis of noble metal nanosheets.

## Discussion
In summary, we report a facile method to engineer the in-plane strain in ultrathin Ir nanosheets by constructing high-density amorphous–crystalline phase boundaries. Geometric-phase analysis and electron diffraction profile reveal that the amorphous–crystalline boundary can generate about 4% tensile strain in the nanosheets. As demonstrated by synchrotron-based valence band spectra and density functional theory investigation, the tensile strain could enable the modification of d-band state of Ir sites and optimize the Gibbs free energy of hydrogen adsorption. As a result, the as-prepared Ir nanosheets exhibit high TOF value (3.61 $H_2$ $s^{-1}$ @ 30 mV), which is 4.5-times higher than that of the benchmark commercial Pt/C catalysts (0.81 $H_2$ $s^{-1}$) towards HER performance. More impressively, the in-plane strain engineering induced by amorphous–crystalline boundaries is also demonstrated to be feasible for boosting the HER catalysis of Ru nanosheets and Rh nanosheets. Therefore, engineering the in-plane strain by constructing amorphous–crystalline phase boundaries offers an

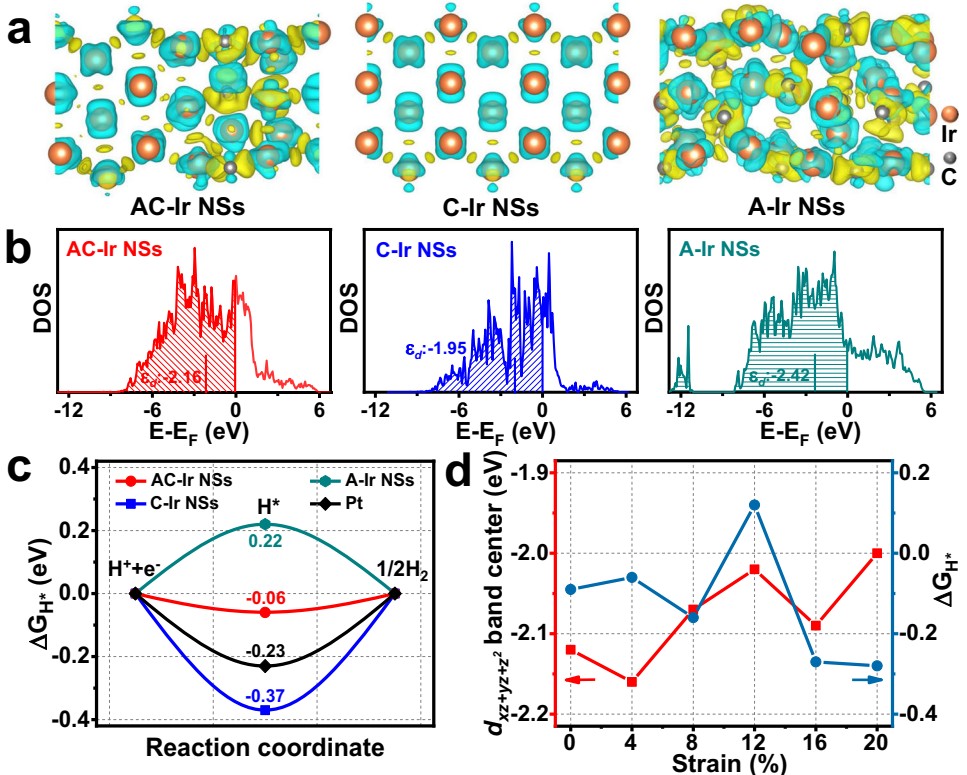

**Fig. 5 Understanding the effect of tensile strain. a** The deformation charge density analysis of AC-Ir NSs with 4% tensile strain, C-Ir NSs and A-Ir NSs, respectively (isosurface is 0.12 bohr$^{-3}$, cyan represent charge depletion and yellow represent charge accumulation.). **b** The $d_{xz+yz+z^2}$ band centers of the active Ir sites for AC-Ir NSs with 4% tensile strain, C-Ir NSs and A-Ir NSs, respectively. Note: $\varepsilon_d$ refers to $E_{dxz+yz+z^2}$. **c** The calculated $\Delta G_{H^*}$ on AC-Ir NSs with 4% tensile strain, C-Ir NSs and A-Ir NSs respectively. **d** The $d_{xz+yz+z^2}$ band centers of the active Ir sites and the calculated $\Delta G_{H^*}$ on AC-Ir NSs with different degrees of lattice expansion.

efficient avenue to the exploration and design of high-performance noble metal nanomaterials for energy-driven applications.

## Methods

**Chemicals**. Iridium 2,4-pentanedionate (Ir(acac)$_3$), Ruthenium acetylacetonate (Ru(acac)$_3$) and Rhodium 2,4-pentanedionate (Rh(acac)$_3$) were purchased from Alfa Aesar. Potassium nitrate (KNO$_3$), Potassium Bromide (KBr) and alcohol were obtained from Sinopharm Chemical Reagent Co., Ltd (Shanghai, China). Deionized (DI) water from Milli-Q System (Millipore, Billerica, MA) was used in all our experiments. 5 wt% Nafion solution and Commercial Pt/C catalysts was purchased from Aldrich. All chemicals were used as received without further purification.

**Synthesis of AC-Ir NSs**. In a typical synthesis of AC-Ir NSs, 10 mg Ir(acac)$_3$ and 20 mg KBr were dissolved into 7 mL ethanol-DI water mixture under stirring. Then, the mixture solution was dried to obtain powder. The obtained powder was heated to 300 ºC for 90 min at the heating rate of 5 ºC/min under air in a tube furnace and then naturally cooled to room temperature. The as-obtained products were washed with DI water-ethanol mixture for several times.

**Synthesis of C-Ir NSs and A-Ir NSs**. The C-Ir NSs were obtained using the same process as that of AC-Ir NSs, except that air was replaced with argon/ hydrogen (v/v = 90/10). Similarly, the A-Ir NSs were obtained using the same process as that of AC-Ir NSs, except that KBr was replaced with KNO$_3$.

**Synthesis of A-Ru NSs, C-Ru NSs, AC-Ru NSs, A-Rh, C-Rh NSs and AC-Rh NSs**. In a typical synthesis of A-Ru NSs (A-Rh NSs), 10 mg Ru(acac)$_3$ (Rh(acac)$_3$) and 20 mg KBr were dissolved into 7 mL ethanol-DI water mixture under stirring. The mixture solution was dried to obtain powder. The obtained powder was heated to 280 ºC for 90 min with the heating rate of 5 ºC/min under air and then naturally cooled to room temperature to obtain the samples. Furthermore, the C-Ru NSs (C-Rh NSs) were obtained using the same process as that of A-Ru NSs (A-Rh NSs), except that air was replaced with argon/hydrogen (v/v = 90/10) and annealing temperature from 280 ºC change to 400 ºC. Furthermore, for the synthesis of AC-Ru NSs (AC-Rh NSs), 5 mg A-Ru NSs (A-Rh NSs) was heated to 320 ºC for 90 min

under argon and then naturally cooled to room temperature. The as-obtained products were washed with DI water-ethanol mixture for several times.

**Characterization**. TEM images of samples were carried out by Hitachi H-7700. SEM images and EDS spectra of samples were recorded on Genimi SEM 500. Aberration-corrected HAADF-STEM images of samples were conducted on JEOL JEM-2010 LaB$_6$ high-resolution transmission electron microscope. XRD patterns of samples were obtained by Rigaku Miniflex-600. Electron diffraction profiles are acquired by rotationally averaging integration of electron diffraction pattern using the PASAD plugin in Digital Micrograph. X-ray photoelectron spectroscopy (XPS) spectrum of samples was assessed at beamline BL10B of National Synchrotron Radiation Laboratory (NSRL) of China using Mg Kα (hν = 1248.6 eV) radiation source. AFM image of AC-Ir NSs was conducted on Dimension ICON with Nano Scope V controller (Bruker). The valence band spectra of samples were collected at beamline BL10B of National Synchrotron Radiation Laboratory (NSRL) of China using synchrotron photo radiation source (hν = 170 eV). The residual pressure in the spectrometer analysis chamber was ~10$^{-9}$ to 10$^{-10}$ mbar. The samples were fixed on a sample holder using a conductive double sided adhesive carbon tape and transferred in the spectrometer analysis chamber, and then vacuumized treatment overnight. The resolving power of the grating was typically E/ΔE = 1000 and the photon flux was 1*10$^{10}$ photons per second. The valence band spectra were energy calibrated by measuring a gold plate in electric contact with the sample and setting the Au 4$f_{7/2}$ core level peak to 84.0 eV. The XAFS data (Ir L$_3$-edge) of samples were collected at BL14W1 beamline of Shanghai Synchrotron Radiation Facility (SSRF). The obtained XAFS data were analyzed by IFEFFIT software packages and wavelet transform analysis was employed using the Igor pro script. The Morlet wavelet was chosen as basis mother wavelet and the parameters (η = 8, σ = 1) were used for a better resolution in the wave vector k.

**Strain analysis around amorphous–crystalline boundary**. Traditional geometric-phase analysis method is used for preliminary strain calculation, which is based on the reciprocal space of HRTEM image and FRWRtools plugin in DigitalMicro-graph software.

**Electrochemical measurements for HER**. All the electrochemical experiments were performed on the CHI 760E electrochemical workstation (Shanghai Chenhua,

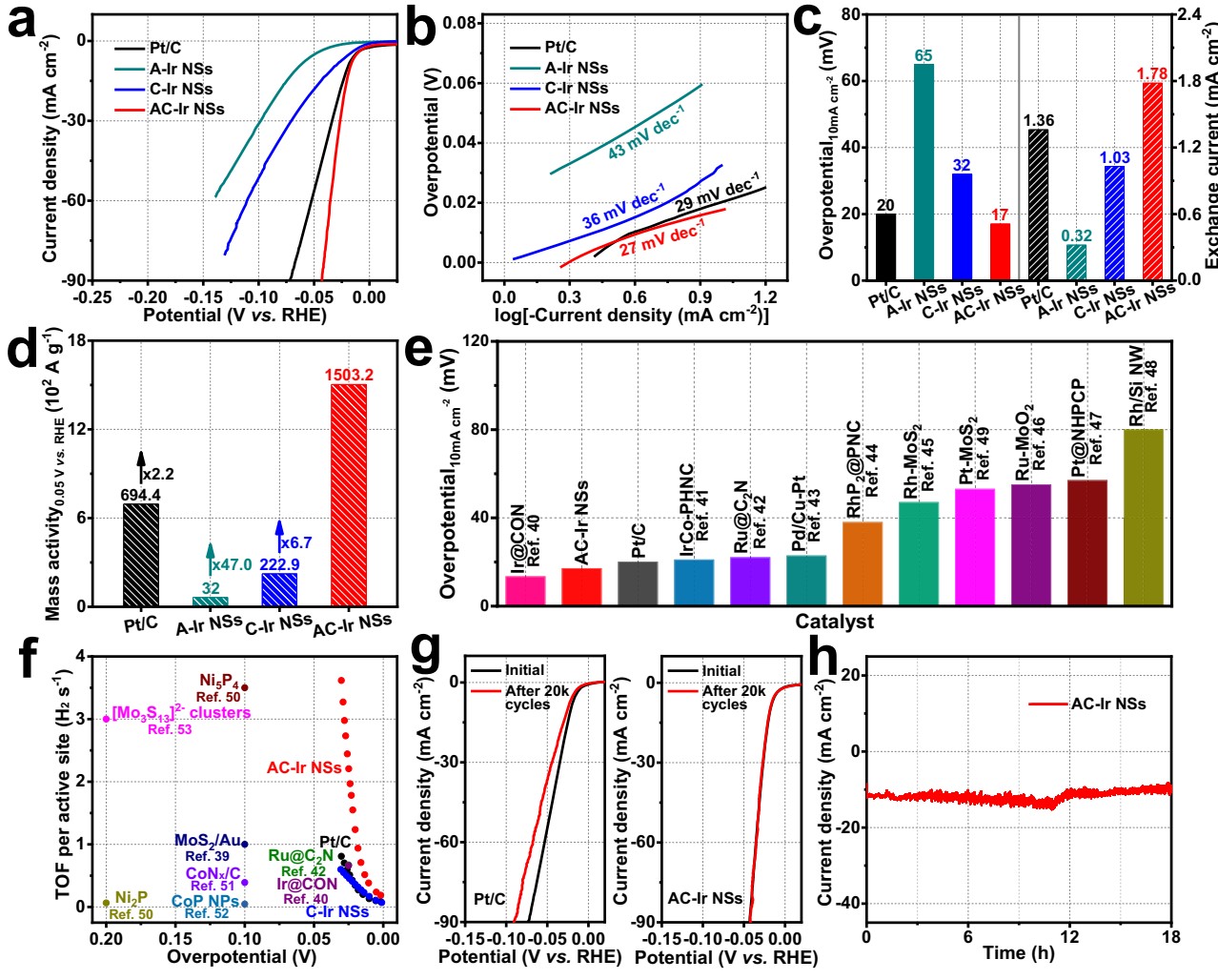

**Fig. 6 Electrochemical HER performance of AC-Ir NSs. a** Polarization curves, (**b**) The corresponding Tafel plots, (**c**) Comparison of the overpotential at 10 mA cm$^{-2}$ (left axis) and exchange current densities (right axis) and (**d**) Mass activity at an overpotential of 50 mV (vs. RHE) of AC-Ir NSs, C-Ir NSs, A-Ir NSs and Pt/C, respectively. **e** Comparison of the overpotential at 10 mA cm$^{-2}$ for AC-Ir NSs, Pt/C and recently reported HER catalysts[40-49]. **f** TOF values of AC-Ir NSs, C-Ir NSs, Pt/C and recently reported HER catalysts[39,40,42,50-53]. **g** Polarization curves of AC-Ir NSs before and after 20,000 cycles and (**h**) Chronoamperometry measurement at an overpotential of 17 mV (vs. RHE) of AC-Ir NSs.

China). Catalysts were tested on the glassy carbon electrode (0.196 cm$^2$) as the working electrode, while graphite rod as the counter electrode and Ag/AgCl as the reference electrodes. The catalyst solutions were prepared by mixing catalysts with Cabot Vulcan XC-72 carbon black (Nobel metal loading: 0.8 mg mL$^{-1}$) in a solution containing 980 µL of ethanol-DI water mixture and 20 µL of Nafion solution by sonication to form homogeneous catalyst inks. Then, 20 µL catalyst ink was dropped onto the glassy carbon rotating disk electrode (RDE) with drying naturally for measurement.

Hydrogen evolution reaction (HER) tests were carried out in N$_2$-saturated 0.5 M H$_2$SO$_4$ solution (acidic medium) (or 1.0 M KOH solution (alkaline medium)) with a scan rate of 5 mV/s at 1600 rpm under room temperature. All potentials were referenced to a reversible hydrogen electrode (RHE) with IR correction where the R was referred to the ohmic resistance. The ohmic resistance was measured by Electrochemical impedance spectroscopy (EIS), where EIS measurements were tested from 100 kHz to 0.1 Hz. The accelerated durability test (ADT) of the catalysts were carried out in N$_2$-saturated 0.5 M H$_2$SO$_4$ solution by applying potential cycling between −0.15 V and −0.45 V (vs. Ag/AgCl) with a scan rate of 50 mV/s for 20,000 cycles under room temperature.

**Active sites calculation**. For CO stripping test, to ensure the saturated adsorption of CO at the surface of the catalyst, CO gas was slowly bubbled into 0.5 M H$_2$SO$_4$ solution for 10 min while maintaining the potential of the working electrode at 0.2 V (vs. RHE). Then, the electrode was transferred to another cell filled with Ar-saturated 0.5 M H$_2$SO$_4$ solution. Then cyclic voltammograms were conducted from 0.05 V to 1.2 V (vs. RHE) with a scan rate of 10 mV s$^{-1}$.

CO stripping test was used to calculate the active sites of the Pt-based (or Ir-based) electrocatalysts. Therefore, the number of active sites (n) can be calculated based on CO stripping test using the following equation.

$$n = \frac{Q_{CO\ stripping}}{2F} \quad (1)$$

where F is the Faraday constant (C mol$^{-1}$).

**Turnover frequency calculation**. The turnover frequency calculation (TOF) was calculated based on Eq. 2 shown as follows:

$$TOF = \frac{I}{2Fn} \quad (2)$$

where I is the current during polarization curves measurement, F is the Faraday constant (C mol$^{-1}$), n is the number of active sites (mol). The factor 1/2 is based on the assumption that two electrons are necessary to form on hydrogen molecule.

**Theoretical calculation methods**. The Density functional theory (DFT) calculations were performed using the Vienna ab initio simulation package (VASP). The Perdew−Burke−Ernzerhof (PBE) form of the generalized gradient approximation (GGA) was used to describe the exchange-correlation functional[54]. Dispersion effects have been accounted for according to Grimme's correction scheme (DFT-D3)[55]. Projector augmented wave (PAW) method was employed to describe the interaction between valence electrons and ionic cores[56]. The kinetic energy cutoff of 450 eV with spin-polarization was utilized to expand the wave function of the

valance electronic states. The van der Waals correction was considered here using the empirical correction in Grimme's scheme. The Monkhorst–Pack $k$-point meshes of $3 \times 3 \times 1$ and $6 \times 6 \times 1$ were used for the Brillouin zone sampling of surface and PDOS calculation, respectively. The convergence tolerances of the optimized calculation were defined as follows: 0.02 eV/Å for the maximum force and $1 \times 10^{-4}$ eV/atom for the SCF calculation.

For the slab model of crystalline structure, the crystalline Ir $(0 -1 1)$ surface is represented by a periodic six-layer slab using a $3 \times 2$ surface unit cell containing 72 Ir atoms (six layers atoms, 12 atoms per layer). The Ir$(0 -1 1)$ surface plane was chosen because it is perpendicular to the $(2 0 0)$ and $(1 1 1)$ planes, which coincides well with the FFT patterns of crystalline structure in Fig. 1. As shown in Supplementary Fig. 15, the models of A-Ir NSs and AC-Ir NSs are built by doping C at the site of Ir vacancy. Concretely, to construct A-Ir NSs model, we replace 24 Ir atoms by C and the ratio of C:Ir is 24:48. For the I, III, IV and V layers, Ir and C atoms are arranged one by one. The II and VI layer are all Ir atoms. To construct AC-Ir NSs model, we replace 8 Ir atoms by C and the ratio of C:Ir is 8:64. Ir and C atoms are arranged one by one in the I, III, V layers of the fifth column, and IV layer of the sixth column. All the others are Ir atoms. For AC-Ir NSs, we simulate the crystalline domain and amorphous domain from left to right along x axis and C atoms are listed around amorphous–crystalline boundary. Besides, to study the strain effect, we apply the tensile strain in the x direction with different ratio for AC-Ir NSs. The surface is modelled by a periodic six-layer slab repeated in $3 \times 2$ surface unit cell with a vacuum region of 12 Å between the slabs along the z axis. Here all the atoms are relaxed for optimization. Furthermore, the coordination numbers of optimized model structures for AC-Ir NSs and A-Ir NSs are close with the results of XAFS analysis. The density difference is the deformation charge density following the formula $\rho = \rho_{sur} - \sum_{i=1}^{m} \rho Ir_m - \sum_{i=1}^{n} \rho C_n$. Here $\rho_{sur}$ is the charge density of surface. $\sum_{i=1}^{m} \rho Ir_m$ and $\sum_{i=1}^{n} \rho C_n$ are the sum of charge density of free iridium atom and carbon atom, respectively. It is calculated by subtracting the charge density of single atom from the charge density of the whole surface. The $d_{xz+yz+z^2}$ band center is calculated by the formula $E = \frac{\int\_y x dx}{\int\_y dx}$, where y is the density of states, x is the energy. The $\Delta G_{H^*}$ is calculated as follow: $\Delta G_{H^*} = E_{(sur+H^*)} - E_{sur} - 1/2 E_{H2} + \Delta E_{ZPE} - T\Delta S_H$[38]. Here, $\Delta E_{ZPE}$ is the difference in zero point energy between the adsorbed and the gas phase, the values are 0.17 eV and 0.135 eV, respectively. The contribution from the vibrational frequencies is negligibly small with harmonic approximation and $\Delta E_{ZPE}$ is 0.04 eV here. $\Delta S_H \approx -1/2 \Delta S^0_{H_2}$ due to the negligible vibrational entropy in the adsorbed state, where $S^0_{H_2}$ is the entropy of $H_2$ in the gas phase at standard conditions. At the temperature of 300 K, $T\Delta S_H$ is calculated to be $-0.2$ eV. As a result, we use 0.24 eV to represent $\Delta E_{ZPE} - T\Delta S_H$. The $E_{(sur+H^*)}$ is the energy of surfaces with $H^*$ adsorption. The $E_{H2}$ and $E_{sur}$ are the energies of individual $H_2$ molecule and catalytic surface.

## Data availability

The data that support the findings of this study are available from the corresponding authors upon reasonable request.

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

## Acknowledgements

This work was supported by the National Key R&D Program of China [2017YFA0700104 (X.H.), 2018YFA0702001 (X.H.), 2017YFA0206703 (G.W.)]; National Natural Science Foundation of China [21871238 (X.H.), 22175163 (G.W.)]; Youth Innovation Promotion Association of the Chinese Academy of Science (2018494; X.H.) and China Postdoctoral Science Foundation (2021M693064; G.W.). We thank the photoemission end stations MCD-A, MCD-B, BL10B in National Synchrotron Radiation Laboratory (NSRL) and BL14W1 in Shanghai Synchrotron Radiation Facility (SSRF) for help in characterizations.

## Author contributions

X.H. and G.W. conceived the idea and co-wrote the paper. G.W. carried out the sample synthesis, characterization and electrocatalytic measurement. X.H. performed the strain analysis. J.C. calculated the DFT calculation. X.Z. and P.C. performed the XAFS characterization. H.L. carried out the AFM characterization. P.Y. and C.C. discussed with results and helped with the modification of the paper.

## Competing interests

The authors declare no competing interests.
