## [Peer Review File · Nature Communications]

In-plane Strain Engineering in Ultrathin Noble Metal Nanosheets Boosts the Intrinsic Electrocatalytic Hydrogen Evolution ActivityEditorial Note: Parts of this Peer Review File have been redacted as indicated to remove third-party material where no permission to publish could be obtained.

Reviewers' comments:

Reviewer #1 (Remarks to the Author):

In this work, the authors use a variety of materials characterization techniques to examine the structure and electrochemical activity of Ir nanosheets. The structure of the material appears highly disordered, which makes it very interesting as a strain-engineered electrocatalyst. I have a large number of concerns with the data quality and the conclusions drawn from this paper. While I appreciate the space limitations of this journal, the conclusions are not currently supported by the data presented, and the cursory discussion of the very large number of results does not produce a compelling analysis. I cannot recommend the publication of this manuscript without significant

The introduction is very light and doesn't really engage with strain-engineered catalysts. There are several reviews on this topic.

It is not obvious to me which synthetic condition (temperature, etc) produces the material shown in each figure. For example, several materials are shown in Figure S1, but which of these does Figure 1 2 and 3 correspond to?

The tensile strain of 25% reported in Figure 2 using GPA is incredibly large. The biggest drawback associated with GPA is that one can only probe the average location of a column of atoms. For strained or misoriented systems, the average position seen in projected view does not correspond meaningfully to the local interatomic distances.

Better XRD results would dramatically enhance the quality of the manuscript, and convince me the strain observed with TEM is representative of the sample. The use of diffraction to probe defect and strain structure in electrocatalysts is quite well developed (for example, DOI: 10.1038/s41563-018-0133-2). The laboratory XRD results do not currently contribute a lot of value, since no signal can be observed. If the average Ir-Ir bond length is substantially lengthened by tensile strain, as argued by the TEM and EXAFS data, the peaks in the XRD should be correspondingly shifted to shorter scattering angles. This analysis is required at a minimum.

I find the discussion of the EXAFS/XPS data also confusing. If a large fraction of the Ir is oxidized (as indicated by XPS), why is the EXAFS peak in Figure B/C interpreted as Ir-C? Shouldn't this be the oxide? In the text, this bond length is attributed to the amorphous Ir and/or grain boundaries. I don't understand how this is possible. The amorphous Ir should have very similar Ir-Ir bond length as the crystalline Ir, but with no long range order. If a huge percentage of the material is actually iridium carbide, then we need much more detailed structural information on exactly what this material is. More discussion of the wavelet EXAFS analysis would be required to differentiate Ir-C and Ir-O, which are very similar indeed.

It is well-known that iridium oxide is often amorphous. Couldn't the amorphous regions detected in the TEM simply correspond to iridium oxide? If so, the conclusions from TEM GPA analysis and the single largest claim made by this manuscript are invalid, but the XPS and EXAFS results make sense. One of the key data points, the Ir-Ir bond length seen in k3-EXAFS, could support the rest of the analysis but seems to not be discussed at all.

The EDS shows that the material is about 13 wt% O (ignoring the C and Si). Pure IrO₂ is about 14 wt% O (32/224 g/mol).

All of this could be corrected by showing TEM chemical analysis of these alleged amorphous crystalline grain boundaries. High resolution EELS maps would convincingly show that the disordered domains are indeed amorphous iridium. Strangely the oxygen EDS maps are not shown alongside the Ir and C

maps, even though the EDS spectra shown in Figure S4 indicates there is much more oxygen than carbon. In any case these maps (Fig 1C) are too low resolution to distinguish the grains being examined.

The valence band spectroscopy is both very interesting, but insufficiently explained. The insufficient characterization of these different samples does not allow a meaningful comparison of their VBS. Data collection and interpretation of these features is exceedingly complex, and requires more than one sentence conclusion. If these results are included, information supporting the surface cleanliness, energy calibration, and analysis should be included. Right now I see no information in the text regarding these measurements, but infer it was performed on the same laboratory XPS instrument used for Figure 3A.

Without more information as to how the theoretical structures of each material were obtained for DFT, I see no way of evaluating the quality of the electronic analysis. Furthermore, these structures seem directly at odds with the XPS/EDS/EXAFS results indicating large amounts of C and O, and a highly oxidized surface.

It is well-known that H-UPD and CO stripping on Ir surfaces is complicated by surface oxidation, which makes the surface area and active site determination shown in Figure S23 unreliable and underestimated. This undermines much of the electrocatalytic work, although mass activity and tafel slopes should be unaffected.

An accelerated stability test for HER is shown in Fig 5g. 20000 cycles are claimed, but I could not find an actual description of between which potentials, scan rates, or any other necessary information. Without cyclic voltammetry data before and after the test showing changes in the Pt surface area, I am not able to determine whether the Ir catalyst is more stable than the Pt catalyst, or whether the Pt catalyst is more easily poisoned by contaminants over time.

Reviewer #2 (Remarks to the Author):

The authors reported a new methods for creating in-plane amorphous-crystalline boundaries, and proved the boundaries generated up to about 25% surface tensile strain. Due to the tensile strain, the as-prepared ultrathin Ir nanosheets showed enhanced intrinsic activity toward HER. I can not recommend this work to be published in Nat. Commun..

1. The authors have reported an approach for amorphous Ir nanosheets towards OER before (Nature Communication, 2019, 10, 4855). Why the difference between KBr and KNO₃ causes such big change in crystal structure?
2. The authors claim that the tensile strain enables the modification of d band state of Ir sites and optimize the Gibbs free energy of hydrogen adsorption. However, in the previous literature reported by authors (Nature Communication, 2019, 10, 4855), the Ir-Ir bond of A-Ir NS (2.92 Å) is also larger than that of Ir power (2.71 Å), indicating the tensile strain in A-Ir NS. However, the HER activity of A-Ir NS is worse than that of AC-Ir NS. Why tensile strain leads to diverse HER performance? The author should provide more evidence why such big tensile strain can contribute to the enhanced HER activity.
3. Considering the much higher price of Ir than that of Pt, it seems non-ideal to replace Pt with Ir in acid media for HER. How about the catalytic performance of AC-Ir NS in alkaline electrolyte?
4. It should be not stable structure if you consider such big tensile strain in nanosheets.

Reviewer #3 (Remarks to the Author):

The manuscript presents a novel catalyst more active for HER than Pt, based on straining Ir. This is a theoretically predicted result, and in itself very interesting. The manuscript goes to great length to argue that the electrocatalytic activity is due to the strain effect. The arguments for this seem pretty solid, given the experimental evidence.

A significant amount of carbon and oxygen is observed in the sample, and this will also have a dramatic effect on HER activity, possibly even larger than the strain effect. That the observed increased activity does not originate from impurities can, however, not entirely be excluded (Sup figure 4).

The DFT calculations need some further explanation in order to be reproducible. It would be important to know what code was used, what exchange-correlation functional, what part of the structures were fixed, which adsorbate structure corresponds to which color line in Sup. fig. 16, what initial structures/adsorption structures were selected/tested and how was minimum energy structures identified, etc.

The manuscript is otherwise excellent, and well worth reading/publishing.

Point-by-point response to the referees' comments

We sincerely thank the referees for their careful review and valuable comments, which certainly help improve the manuscript. We also appreciate the offered opportunity to address the comments and revise our manuscript. Our point-by-point responses are presented below and all the changes in the revised manuscript have been highlighted in red for your review.

Response to the Reviewer 1:

In this work, the authors use a variety of materials characterization techniques to examine the structure and electrochemical activity of Ir nanosheets. The structure of the material appears highly disordered, which makes it very interesting as a strain-engineered electrocatalyst. I have a large number of concerns with the data quality and the conclusions drawn from this paper. While I appreciate the space limitations of this journal, the conclusions are not currently supported by the data presented, and the cursory discussion of the very large number of results does not produce a compelling analysis. I cannot recommend the publication of this manuscript without significant revisions.

Response: We warmly appreciate the valuable comments and suggestions, and also thank the referee for recommending significant revision. Following the suggestions, we have performed more characterizations, such as high-resolution EELS spectroscopy, the electron diffraction profile and X-ray diffraction, and provide refined data analysis and discussion on EXAFS spectrum, oscillation curves, and wavelet transform spectra to improve the manuscript. Details are listed below.

Comment 1-1: The introduction is very light and doesn't really engage with strain-engineered catalysts. There are several reviews on this topic.

Response: We thank the referee for the valuable suggestions. To enhance the comprehensiveness, we have revised the introduction and provide more discussion on the advancement of the strain-engineered catalysts and the advantages of the designed in-plane strain engineering. The added discussion are as follows: "*since surface strain can modulate the electronic structure of catalysts by shifting the d-band centre, which is closely related with the adsorption energy and activation energy barriers of reaction intermediates during catalytic process* (Page 2, line 52)" "*For example, the*

intermetallic PtPb core can induce conformal four-layer Pt shell with tensile strain of 7.5% in PtPb/Pt core-shell nanoplate, which enable the catalyst with optimal oxygen adsorption energies and thus superior performance of oxygen reduction reaction. Besides, the strain of ultrasmall Pt nanoparticles supported on LiCoO₂ can be tuned from compressive to tensile by altering lattice spacing of LiCoO₂ substrate and further optimize their catalytic performance. (Page 3, line 64)” and “the strain induced by lattice mismatch is spatially dependent, which generally decays from the interface toward the outmost surface and even vanishes beyond several atomic layers. Therefore, the catalytic activity is highly related to the thickness of shell in core-shell structure or the size of metal in metal-substrate structure. (Page 3, line 66)”.

Furthermore, to clearly reveal advantages of in-plane strain engineering, we also add schematic illustration of strain constructing through lattice mismatch approach and in-plane strain engineering in the revised manuscript. Meanwhile, more literature and reviews about strain-engineered catalysts were cited in the revised manuscript (*Nature* 2021, 598, 76-81; *Adv. Mater.* 2019, 31, 1807001; *Chem. Soc. Rev.* 2018, 47, 3100-3128.). The revision is highlighted in the revised manuscript.

Scheme 1. Schematic illustration of strain constructing through the lattice mismatch approach (core-shell structure and metal-substrate structure) and the proposed in-plane strain engineering.

Comment 1-2: It is not obvious to me which synthetic condition (temperature, etc) produces the material shown in each figure. For example, several materials are shown in Figure S1, but which of these does Figure 1 2 and 3 correspond to?

Response: We thank the referee for carefully reviewing our manuscript and we are pleased to clarify the difference. Materials of S1a, S1b and S1c correspond to amorphous Ir nanosheets, AC- Ir nanosheets and crystalline Ir nanosheets, which were fabricated at 280 °C, 300 °C and 360 °C, respectively. The material in Figure 1, 2, 3 is AC-Ir nanosheets, obtained at 300 °C. To make it clear for readers, we have revised the manuscript and added the synthetic conditions in the figure captions.

Comment 1-3: The tensile strain of 25% reported in Figure 2 using GPA is incredibly large. The biggest drawback associated with GPA is that one can only probe the average location of a column of atoms. For strained or misoriented systems, the average position seen in projected view does not correspond meaningfully to the local interatomic distances.

Response: We appreciate the constructive comments. We agree with the reviewer's viewpoint that GPA analysis has drawback towards strained or misoriented systems. However, for crystalline structure, GPA analysis can effectively assess their strained distribution based on HRTEM image where the direction of incident electron beam is parallel to zone axis. For amorphous structure, although the absolute value of strain obtained by GPA is not precise enough, GPA analysis may give an overall trend about the strain distribution in the structure. We used GPA to analyze the strain of crystalline domain in the AC-Ir nanosheet (Figure R1). The maximum strain of 25% is mainly existent in amorphous domain. Considering the strain of amorphous domain obtained by GPA maybe not precise enough and the value of strain in crystalline domain is mainly concentrated on 4%, it could be reasonable to express the strain of nanosheets using the average value of 4%.

Besides, we also employed electron diffraction profile (EDP) acquired by rotationally averaging integration of electron diffraction pattern, to probe the strain of heterostructured amorphous-crystalline materials (*Sci. Adv.* 2019, 5, eaau9590.). As revealed in Figure R2 and Table R1, the diffraction peaks of AC-Ir NSs show notable negative shifts compared with Ir crystals. According to EDP analysis, the lattice expansion of AC-Ir NSs is estimated to be around 4%, which is consistent to the strain of the crystalline domains obtained by GPA. In addition, the diffraction peaks observed in the XRD shifts negatively (Figure 3), suggesting the existence of stain tensile in the materials.

Considering the accuracy of the GPA analysis, we removed the description about

maximum 25% strain in the revised manuscript, and used the average strain of 4% in the revised manuscript. Related description and discussion were highlighted in the revised manuscript (Page 6, Line 140).

Figure R1. (a) HRTEM image of amorphous-crystalline boundary. (b) e_{yy} strain component acquired by GPA method. (c) The histogram of strain distribution.

Figure R2. Electron diffraction profiles of AC-Ir NSs. (a) Electron diffraction profile of AC-Ir NSs. (b) and (d) Selected area electron diffraction patterns (SAED) of Ir Crystal and AC-Ir NSs, respectively. (c) and (e) Selected area (white dotted circle) of SAED for Ir Crystal and AC-Ir NSs, respectively.

Table R1. The average strain of AC-Ir NSs obtained by Electron diffraction profile.

AC-Ir NSs		Ir Crystal		Strain
d^{-1} (1/nm)	d (nm)	d^{-1} (1/nm)	d (nm)	
4.40	0.23	4.54	0.22	~4%

Comment 1-4: Better XRD results would dramatically enhance the quality of the manuscript, and convince me the strain observed with TEM is representative of the sample. The use of diffraction to probe defect and strain structure in electrocatalysts is quite well developed (for example, DOI: 10.1038/s41563-018-0133-2). The laboratory

XRD results do not currently contribute a lot of value, since no signal can be observed. If the average Ir-Ir bond length is substantially lengthened by tensile strain, as argued by the TEM and EXAFS data, the peaks in the XRD should be correspondingly shifted to shorter scattering angles. This analysis is required at a minimum.

Response: We thank the reviewer for the valuable suggestions. We agree that the diffraction obtained by synchrotron wide-angle X-ray scattering (WAXS) measurements can precisely explore defect and strain structure in materials. Unfortunately, due to the lack of beam time in the scattering beamline, the synchrotron WAXS data cannot be obtained. To solve this issue, we used slow step measurement to obtain better diffraction pattern (Measurement condition: step: 0.01 degree, scan speed: 1.0 degree min⁻¹), as shown in Figure R3. As expected, the diffraction peaks of AC-Ir NSs show negative shifts relative to the Ir crystal, confirming the tensile strain can be generated in AC-Ir NSs. According to Bragg's Law ($2d\sin\theta=n\lambda$, $\lambda_{\text{Cu}}=0.15406$ nm), the lattice spacing of different plane in AC-Ir NSs and Ir crystal were obtained and shown in Table R2. Compared with Ir crystal, the lattice spacing of all lattice planes in AC-Ir NSs were increased about 2%. Since the lattice expansion is positively correlated with surface strain, the lattice expansion of 2% also reveal AC-Ir NSs possess tensile strain. Combining the data of XRD, GPA and electron diffraction profile, it is convincing to conclude amorphous-crystalline boundary can introduce lattice strain in the two-dimensional nanosheet.

Figure R3. XRD patterns of AC-Ir NSs and Ir nanocrystal.

Table R2. The lattice spacing of different planes in AC-Ir NSs and Ir crystal obtained by XRD.

Lattice plane (h k l)	AC-Ir NSs		Ir Crystal		Increasement
	2 Theta (°)	d (nm)	2 Theta (°)	d (nm)	
(111)	40.1	0.225	40.7	0.221	~2%
(200)	46.5	0.195	47.3	0.192	~2%
(220)	68.4	0.138	69.1	0.135	~2%

Comment 1-5: I find the discussion of the EXAFS/XPS data also confusing. If a large fraction of the Ir is oxidized (as indicated by XPS), why is the EXAFS peak in Figure B/C interpreted as Ir-C? Shouldn't this be the oxide? In the text, this bond length is attributed to the amorphous Ir and/or grain boundaries. I don't understand how this is possible. The amorphous Ir should have very similar Ir-Ir bond length as the crystalline Ir, but with no long-range order. If a huge percentage of the material is actually iridium carbide, then we need much more detailed structural information on exactly what this material is. More discussion of the wavelet EXAFS analysis would be required to differentiate Ir-C and Ir-O, which are very similar indeed.

Response: We thank the referee for the valuable comments and we are pleased to clarify this issue. The assignment of the oxidized state of Ir to the Ir-C bond has been carefully considered. The reasons are as follows.

First, the XPS Ir 4f spectrum of AC-Ir NSs are totally different from that of IrO₂, as shown in Figure R4f. The fitted peaks of the oxidized state of Ir in AC-Ir NSs are located at 62.2 and 65.0 eV, which are negatively shifted in comparison with IrO₂ (62.5 eV and 65.4 eV), suggesting the chemical states of Ir in AC Ir NSs are different from Ir-O bonds in IrO₂. In addition, the high-resolution EELS spectroscopy of AC-Ir NSs can probe the element distribution of amorphous and crystalline domain in AC-Ir NSs. As shown in Figure R5, the signal of C is mainly observed in the amorphous domains, while C signal is almost absent in the crystalline domains. The O signal is very limited in both amorphous domains and crystalline domains. Combing these data, we reasonably assign the oxidized Ir in AC Ir NSs to Ir-C.

Figure R4. Ir 4f XPS spectrum of AC-Ir NSs and commercial IrO₂.

Figure R5. (a) Aberration-corrected HAADF-STEM image around amorphous-crystalline phase boundary (inset: dark cyan: amorphous domain, dark green: crystalline domain.). (b) EELS spectroscopy of amorphous domain and crystalline domain corresponding to the selected area in (a), respectively.

For the Ir-C and Ir-O, the bond length should be very close in principle. To assign the Ir-O and Ir-C in the wavelet EXAFS analysis, we cannot just take it like a black box. We should consider the physiochemical properties of samples obtained by other characterizations. Considering the data of EELS and XPS which did not show obvious oxygen signal and Ir-O interaction, we therefore assign it as Ir-C bond. To precisely analyse the wavelet EXAFS, we also performed wavelet transform spectra (WT) of AC-Ir NSs, As shown in Figure R6 and R7, compared with the intensity at about 7 \AA^{-1} of Ir-O coordination in commercial IrO₂, the intensity maxima at around 6 \AA^{-1} could be attributed to Ir-C coordination, in accordance with the calculated Ir-C path ($k = 6.1 \text{ \AA}^{-1}$). Besides, the intensity maxima around 11 \AA^{-1} are corresponded to Ir-Ir coordination, which resembled the Ir-Ir coordination in Ir powder.

Figure R6. The wavelet transform spectra of AC-Ir NSs, Ir powder and IrO₂, respectively.

Figure R7. Comparison of FEFF-calculated the q -space magnitudes for Ir-C path and Ir-O path, respectively.

Since the C element is mainly present in the amorphous domains, the Ir-C bond in AC-Ir NSs mainly originates from the amorphous region rather than crystalline region. Furthermore, we agree with reviewer that the amorphous Ir should have very similar Ir-Ir bond length as the crystalline Ir. Based on the analysis of wavelet transform spectra, the Ir-Ir coordination in AC-Ir NSs resembles that in Ir powder, indicating the similar atomic configuration for Ir site in both AC-Ir NSs and metallic Ir, which exclude the existence of large percentage of iridium oxide and iridium carbide in the AC-Ir NSs. Combined with EELS spectroscopy, we reasonably believe the amorphous region in AC-Ir NSs could be C-containing amorphous Ir. Due to the C doping, the Ir-Ir bond length of amorphous domain is a little larger than that of crystalline Ir.

Furthermore, related description and discussion about the EELS spectroscopy, XPS spectra and wavelet transform spectra have been added in the revised manuscript.

Comment 1-6: It is well-known that iridium oxide is often amorphous. Couldn't the amorphous regions detected in the TEM simply correspond to iridium oxide? If so, the conclusions from TEM GPA analysis and the single largest claim made by this

manuscript are invalid, but the XPS and EXAFS results make sense. One of the key data points, the Ir-Ir bond length seen in k^3 -EXAFS, could support the rest of the analysis but seems to not be discussed at all.

Response: We thank the referee for the constructive comments. In the response to the comment 1-5, we have employed the high-resolution EELS spectroscopy, XPS and the wavelet transform spectra to conclude the oxidative Ir species are not iridium oxides, but the Ir-C species in the amorphous region.

In addition, following the suggestion, more structural analysis on EXAFS has been conducted. From the Ir L_3 -edge $k^3\chi(k)$ oscillation curves (Figure R8), the oscillation feature of AC-Ir NSs is very close to that of Ir powder and apparently different from that of IrO_2 , indicating the existence of IrO_2 in the AC-Ir NSs is very limited. Moreover, these data coincide with the wavelet transform spectra (Figure R6), in which the Ir-Ir coordination in AC-Ir NSs resembles that in Ir powder. As shown in EXAFS spectrum (Figure R9), the peak appearing at about 2.0 Å is associated with the Ir-C bond. The peak at about 2.9 Å is ascribed to Ir-Ir bond of AC-Ir NSs, which is slightly larger than that of Ir powder (2.7 Å) and far less than that of IrO_2 (3.5 Å) and that of iridium carbide (3.1 Å - 3.5 Å, Table R3).

Based on the analysis of the high-resolution EELS spectroscopy, oscillation curves, EXAFS spectrum and wavelet transform spectra, the amorphous domain in AC-Ir NSs could be C-containing amorphous Ir domains instead of amorphous iridium oxides or iridium carbide. Furthermore, related description and discussion about oscillation curves and EXAFS spectrum were added in the revised manuscript.

Figure R8. The Ir L_3 -edge $k^3\chi(k)$ oscillation curves of AC-Ir NSs, Ir powder and IrO_2 , respectively.

Figure R9. The Fourier transformed of k^3 -weighted $\chi(k)$ -function of the EXAFS spectrum of AC-Ir NSs, Ir powder and IrO_2 , respectively.

Table R3. The Ir-C bond length and Ir-Ir bond length of different iridium carbide. Note: the data source originates from Materials Project, <https://materialsproject.org>.

Shell	IrC	IrC_2	IrC_3	IrC_4	Ir_4C_5
Ir-C	2.00 Å	2.19 Å	2.13 Å	2.20 Å	2.35 Å
Ir-Ir	3.27 Å	3.10 Å	3.49 Å	3.27 Å	3.46 Å

Comment 1-7: The EDS shows that the material is about 13 wt% O (ignoring the C and Si). Pure IrO_2 is about 14 wt% O (32/224 g/mol).

Response: We thank the referee for the valuable comments. We totally understand the referee's concern about oxygen content.

Firstly, the EDS spectra of AC-Ir NSs was recorded on Genimi SEM 500. In detail, the solution of AC-Ir NSs was dropped onto the silicon wafer with drying naturally and then fixed on a sample holder for measurement using a conductive double sided adhesive carbon tape. We have retaken the spectra and found the oxygen signal ratio is quite arbitrary. For this issue, we have discussed with the instrument technician. He said the EDS measurement cannot give quantitative results, because it depends on the sample coverage on the silicon (with native oxide coating). The arbitrary oxygen ratio is probably stemmed from the signal of silicon wafer with native oxide coating, which cannot deliver quantitative information for stoichiometric analysis.

To exclude the interference of silicon, we performed HRTEM-EELS spectra, which is more sensitive to light elements (such as C, O) and is an effective approach to assess the content of C/O especially in local position. (*Ultramicroscopy*, 2004, 101, 207-224). As shown in the high-resolution EELS spectroscopy (Figure R5), it is clear

that the O signal is very limited in both amorphous and crystalline domains, suggesting the possibility of IrO₂ is very low. Meanwhile, the oscillation frequency in the Ir L₃-edge $k^3\chi(k)$ oscillation curves of AC-Ir NSs (Figure R8) is very similar to that of Ir powder, indicating the similar atomic configuration for Ir site in both AC-Ir NSs and metallic Ir. Therefore, we reasonably believe the fabricated samples are not iridium oxides.

Comment 1-8: All of this could be corrected by showing TEM chemical analysis of these alleged amorphous crystalline grain boundaries. High resolution EELS maps would convincingly show that the disordered domains are indeed amorphous iridium. Strangely the oxygen EDS maps are not shown alongside the Ir and C maps, even though the EDS spectra shown in Figure S4 indicates there is much more oxygen than carbon. In any case these maps (Fig 1C) are too low resolution to distinguish the grains being examined.

Response: We thank the reviewer for the valuable suggestions. Following the suggestions, we have conducted the high-resolution EELS spectroscopy of amorphous domain and crystalline domain at multiple locations, respectively (Figure R5 and R10). As known, because of the high fluorescence yield for x-ray production and poorer signal/background in the energy loss spectrum resulting from the low M-shell ionization cross section, the EELS signal of Ir element (Ir M_{4,5}-edge, 2040 - 2116 eV) is insensitive and difficult to be detected through HRTEM-EELS technology (*Microsc. Microanal. Microstruct.* 1991, 2, 231-244; *J. Electron Spectrosc. Relat. Phenom.* 2005, 143, 43-50.). Since EELS is sensitive to light elements (such as C, O) and is an effective approach to assess the content of C/O in local area, the EELS spectroscopy of C K-edge and O K-edge in amorphous domain and crystalline domain were analysed, respectively. Clearly, the signal of C is mainly observed in the amorphous domain and the O signal is very limited in both amorphous domains and crystalline domains, revealing amorphous domain of AC-Ir NSs could be C-containing amorphous iridium.

Figure R5. (a) Aberration-corrected HAADF-STEM image around amorphous-crystalline phase boundary (inset: dark cyan: amorphous domain, dark green: crystalline domain.). (b) EELS spectra of amorphous domain and crystalline domain corresponding to the selected area in (a), respectively.

Figure R10. High-resolution EELS spectroscopy analysis around amorphous-crystalline phase boundary at different areas. (a) Aberration-corrected HAADF-STEM image around amorphous-crystalline phase boundary. (b) EELS spectra of amorphous domain and crystalline domain corresponding to the selected area in (a), respectively.

Furthermore, following the reviewer's suggestion, the oxygen EDS maps have been added (Figure R11, Figure 1c in the revised manuscript). Apparently, the amounts of O elements are very limited, in comparison with carbon and iridium. Since EDS elemental mapping images are too low resolution to distinguish the grains being examined, following the reviewer's suggestion, the aberration-corrected HAADF-STEM image around amorphous-crystalline phase boundary and the EELS spectroscopy of amorphous domain and crystalline domain in AC-Ir NSs have been added in the revised Figure 1.

Figure R11. EDS elemental mapping images of AC-Ir NSs.

Comment 1-9: The valence band spectroscopy is both very interesting, but insufficiently explained. The insufficient characterization of these different samples does not allow a meaningful comparison of their VBS. Data collection and interpretation of these features is exceedingly complex, and requires more than one sentence conclusion. If these results are included, information supporting the surface cleanliness, energy calibration, and analysis should be included. Right now, I see no information in the text regarding these measurements, but infer it was performed on the same laboratory XPS instrument used for Figure 3A.

Response: Thank you for the suggestion. Following the suggestion, the characterization and discussion of valence band was described as follows:

The valence band spectra of samples were collected at beamline BL10B of National Synchrotron Radiation Laboratory (NSRL) of China using synchrotron photo radiation source ($h\nu=170$ eV). The residual pressure in the spectrometer analysis chamber was $\sim 10^{-9}$ - 10^{-10} mbar. The samples were fixed on a sample holder using a conductive double sided adhesive carbon tape and transferred in the spectrometer analysis chamber, and then vacuumized treatment overnight. The resolving power of the grating was typically $E/\Delta E = 1000$ and the photon flux was 1×10^{10} photons per second. The valence band spectra were energy calibrated by measuring a gold plate in electric contact with the sample and setting the Au $4f_{7/2}$ core level peak to 84.0 eV. The description of valence band characterization was added in “Characterization” section.

Meanwhile, more discussion about synchrotron-based valence band spectrum were added in revised manuscript. As in section of “Discussion” (Page 8, Line 191), we have expressed that: “From Fig. 3e, the valence band maximum values are determined to be approximately 0.68 eV, 0.52 eV and 1.45 eV for AC-Ir NSs, C-Ir NSs

and A-Ir NSs, respectively. The metal d-band maximum with respect to the Fermi level (E_F) in AC-Ir NSs show positive shifts relative to the C-Ir NSs without strain, revealing anti-bonding state upshift and less electron filling of metal-H anti-bonding orbital in AC-Ir NSs compared with C-Ir NSs, which illuminates the planar strain can modulate the d-band state of Ir sites in AC-Ir NSs.”

Comment 1-10: Without more information as to how the theoretical structures of each material were obtained for DFT, I see no way of evaluating the quality of the electronic analysis. Furthermore, these structures seem directly at odds with the XPS/EDS/EXAFS results indicating large amounts of C and O, and a highly oxidized surface.

Response: We thank the reviewer for the valuable suggestions. We constructed theoretical structures of each material in our work based on the coordination structural information obtained from XAFS analysis (Table R4). The coordination numbers of theoretical structures for AC-Ir NSs, A-Ir NSs and C-Ir NSs are close with the results of XAFS analysis and as shown in Table R5.

Table R4. EXAFS fitting parameters at the Ir L_3 -edge for Ir powder, A-Ir NSs and AC-Ir NS ($S_0^2=0.807$).

Sample	Shell	N ^a	R(\AA) ^b	$\sigma^2(\text{\AA}^2)$ ^c	$\Delta E_0(\text{eV})$ ^d	R factor
Ir powder	Ir-Ir	12.0	2.71	0.0031	8.3	0.0012
A-Ir NSs	Ir-C	5.1	2.01	0.0018	9.4	0.0013
	Ir-Ir	6.3	2.92	0.0076		
AC-Ir NSs	Ir-C	4.5	2.00	0.0050	9.3	0.0065
	Ir-Ir	7.2	2.90	0.0076		

Note: ^aN: coordination numbers; ^bR: bond distance; ^c σ^2 : Debye-Waller factors; ^d ΔE_0 : the inner potential correction.

Table R5. The coordination numbers of AC-Ir NSs, A-Ir NSs and C-Ir NSs based on theoretical structures.

Theoretical model	Shell	Coordination numbers
C-Ir NSs	Ir-Ir	10.0
A-Ir NSs	Ir-C	4.1
	Ir-Ir	5.8

AC-Ir NSs	Ir-C	4.0
	Ir-Ir	7.3

The details about DFT have been described in “Method” section. *“The DFT calculations were performed using the Vienna ab initio simulation package VASP. Generalized gradient approximation (GGA) in the form of the Perdew–Burke–Ernzerhof (PBE) function was used as the exchange-correlation interactions. PAW-PBE was employed to describe the interaction between valence electrons and ionic core. The kinetic energy cutoff of 450 eV with spin-polarization was utilized to expand the wave function of the valance electronic states. The van der Waals correction was considered here using the empirical correction in Grimme’s scheme. The Monkhorst-Pack k-point meshes of 3×3×1 was used for the Brillouin zone sampling of surface. The convergence tolerances of the optimized calculation were defined as follows: 0.02 eV/Å for the maximum force and 1×10⁻⁴ eV/atom for the SCF calculation.”*

Comment 1-11: It is well-known that H-UPD and CO stripping on Ir surfaces is complicated by surface oxidation, which makes the surface area and active site determination shown in Figure S23 unreliable and underestimated. This undermines much of the electrocatalytic work, although mass activity and tafel slopes should be unaffected.

Response: We agree with the reviewer that H-UPD and CO stripping on Ir surfaces is complicated to determine surface area and active sites by surface oxidation, which might underestimate the surface area. However, the value obtained by CO stripping and H-UPD may qualitatively reflect the trend of surface area and active site number to some extent. Besides, we also employed Cu-UPD method to assess the number of active sites for AC-Ir NSs, as shown in Figure R12. The obtained results are very consistent to the result acquired by CO stripping. In addition, following the valuable suggestions raised by the referee, we also analysed the Tafel slopes and mass activity (Figure R13), which is unaffected by surface area. The AC-Ir NSs possess the higher mass activity (1503.2 A g⁻¹) than C-Ir NSs (222.9 A g⁻¹) and A-Ir NSs (32.0 A g⁻¹), suggesting the intrinsic activity has been substantially improved by the interfacial strain engineering. Meanwhile, the decreased Tafel sloped in the interfaced system also means the boosted reaction kinetics toward HER catalysis. Aiming at the issues of the active site

determination and intrinsic activity evaluation, we also give corresponding discussion on them in the revised manuscript.

Figure R12. (a) CO stripping voltammetry of AC-Ir NSs at scan rate of 10 mV s⁻¹. Stripping of a monolayer of CO in first cycle (red), subsequently second cycle after the stripping of CO (black). (b) Copper UPD in 1.0 M H₂SO₄ in the absence (black curve) and presence (red curve) of 5 mM CuSO₄ on AC-Ir NSs, Red curve: the electrode was polarized at 0.27 V for 100 s to form the UPD layers. Note: Scan rate: 10 mV s⁻¹. (c) Estimation of the active sites of AC-Ir NSs through CO-stripping and Cu-UPD, respectively.

Figure R13. (a) Tafel slopes and (b) mass activity at an overpotential of 50 mV (vs. RHE) of AC-Ir NSs, C-Ir NSs, A-Ir NSs and Pt/C, respectively.

Comment 1-12: An accelerated stability test for HER is shown in Fig 5g. 20000 cycles are claimed, but I could not find an actual description of between which potentials, scan rates, or any other necessary information. Without cyclic voltammetry data before and after the test showing changes in the Pt surface area, I am not able to determine whether the Ir catalyst is more stable than the Pt catalyst, or whether the Pt catalyst is more easily poisoned by contaminants over time.

Response: We thank the reviewer for the valuable suggestions. Following the suggestion, the description about accelerated durability tests (ADT) tests has been added in the revised manuscript in “Characterization” section. we have expressed that: “The accelerated durability test of the catalysts was carried out in N₂-saturated 0.5 M

H₂SO₄ solution by applying potential cycling between -0.15 V and -0.45 V (vs. Ag/AgCl) with a scan rate of 50 mV/s for 20,000 cycles under room temperature.”

Furthermore, as shown in cyclic voltammetry (Figure R14), the electrochemical active surface area (ECSA) of commercial Pt/C catalyst underwent about 20% loss after 20,000 cycles. TEM images indicate the well-dispersed Pt nanoparticles in the commercial Pt/C catalysts agglomerated into irregular larger nanoparticles after ADT, suggesting the instability of commercial Pt/C catalysts during ADT test. Compared with commercial Pt/C catalyst, the ECSA of AC-Ir NSs shows ignorable change after 20,000 cycles (Figure R15), revealing AC-Ir NSs are electrochemically robust.

Figure R14. (a) Cyclic voltammetry curves, (b) ECSA and (c, d) TEM images of commercial Pt/C catalysts before and after ADT. The dotted circles in d correspond to the aggregation of Pt nanoparticles.

Figure R15. The electrochemical active surface area of AC-Ir NSs before and after ADT. (a, b) CO stripping voltammetry of AC-Ir NSs before and after ADT. Stripping of a monolayer of CO in first cycle (red), subsequently second cycle after the stripping of CO (black). (c) Estimation of ECSA of AC-Ir NSs before and after ADT.

Response to the Reviewer 2:

The authors reported a new method for creating in-plane amorphous-crystalline boundaries, and proved the boundaries generated up to about 25% surface tensile strain. Due to the tensile strain, the as-prepared ultrathin Ir nanosheets showed enhanced intrinsic activity toward HER. I can not recommend this work to be published in *Nat. Commun.*

Response: We thank the referee for reviewing our manuscript. We are pleased to justify the novelty and significance of this work. Our point-by-point response to the comments are as follows.

Comment 2-1: The authors have reported an approach for amorphous Ir nanosheets towards OER before (*Nature Communication*, 2019, 10, 4855). Why the difference between KBr and KNO_3 causes such big change in crystal structure?

Response: We thank the referee for carefully reviewing our manuscript and our previous article about amorphous nanosheets (*Nat. Commun.* 2019, 10, 4855.). Actually, these two papers target different scientific questions, as well as different structural design at atomic scale. Even the adopted salt template (KNO_3), it also has quite different roles in structural construction.

From the perspective of synthesis, fabricating ultrathin noble metal nanosheet with in-plane heterojunction and controlled strain engineering is synthetically challenging. Fortunately, we successfully achieve such structures by improving our previous amorphous nanosheet synthesis (*Nat. Commun.* 2019, 10, 4855). Directly using previous method, simply heating cannot generate in-plane amorphous-crystalline interface. In this work, we fortunately found the salt templates play vital effect in forming in-plane interface with controlled strain engineering by controlling the diffusion and nucleation rate. As shown in Table R6, with the increase of annealing temperature, the nucleation rate of Ir enhances, leading to the modulation of Ir nanosheets from amorphous structure to heterojunction structure and final crystalline structure by using KBr template. As the difference in the interaction of Ir with KBr / KNO_3 salt, the Ir atoms might possess lower nucleation rate on KNO_3 surface than that on KBr surface, resulting in still amorphous Ir nanosheets obtained at annealing temperature of 300 °C. When the annealing temperature is above 300 °C, the KNO_3 salt molts and reacts with iridium acetylacetonate, finally leading to the mixture of Ir and

$K_{0.2}IrO_2$ products rather than amorphous-crystalline Ir nanosheets.

From the perspective of fundamental science, constructing in-plane heterojunction with tunable strain tensile to maximally exposing strained sites for catalysis is highly meaningful, because it solves the challenges of traditional strain generation via core-shell nanostructures or metal-substrate interfaces, in which the strained interface is basically embedded and unexposed for catalytic reaction. Meanwhile, we also demonstrate this method is a general approach for Ir, Ru and Rh, suggesting the catalytic generality of the in-plane strain engineering. In summary, this work targets a completely different scientific question using a conceptually novel strategy to tune the hydrogen adsorption behavior for HER catalysis, in comparison with our previous amorphous materials.

Table R6. The products obtained at different reaction temperature and salt template.

Temperature	KBr salt	KNO ₃ salt
280 °C	Amorphous Ir nanosheets	Amorphous Ir nanosheets
300 °C	Amorphous-crystalline Ir nanosheets	Amorphous Ir nanosheets
360 °C	Crystalline Ir nanosheets	The mixture of Ir and $K_{0.2}IrO_2$

Comment 2-2: The authors claim that the tensile strain enables the modification of d band state of Ir sites and optimize the Gibbs free energy of hydrogen adsorption. However, in the previous literature reported by authors (Nature Communication, 2019, 10, 4855), the Ir-Ir bond of A-Ir NS (2.92) is also larger than that of Ir power (2.71), indicating the tensile strain in A-Ir NS. However, the HER activity of A-Ir NS is worse than that of AC-Ir NS. Why tensile strain leads to diverse HER performance? The author should provide more evidence why such big tensile strain can contribute to the enhanced HER activity.

Response: We thank the referee for the constructive comments. The active adsorption site in A-Ir NSs differs from that in AC-Ir NSs for HER catalysis, which could lead to the inferior HER activity of A-Ir NS than that of AC-Ir NS. Furthermore, the poor electroconductivity of A-Ir NSs also affect their HER catalysis. The reasons are as follows.

It has been a controversial scientific question whether the homogeneous amorphous material itself possess strain. A majority of scientists support the view that

the homogeneous amorphous materials don't have strain because of strain relaxation (*Nat. Mater.* 2005, 4, 33-36, *Nat. Commun.* 2018, 9, 982.), although the local bond length might be elongated due to the disordered structures. Actually, the tensile strain we discussed in this manuscript is located in the crystalline domain, which is created by the crystalline-amorphous boundary. The bond length changes in ordered crystalline structures can reflect the strain feature. In order to explore the effects of amorphous-crystalline interface on the bond length of Ir-Ir, we have conducted the radial distribution function (RDF) of A-Ir NSs and AC-Ir NSs. As revealed in Figure R16, the amorphous domain in AC-Ir NSs possess slight larger Ir-Ir bond than A-Ir NSs. Furthermore, following the analysis of GPA (see the response to the comment 1-3 of the first referee), the crystalline domain in AC-Ir NSs possesses the average tensile strain of 4%, revealing amorphous-crystalline interface induce the crystalline domain possess the tensile strain. Besides, DFT calculations were also performed to study the H* adsorption on the strained crystalline domain and the amorphous structures, as shown in Figure R17. Clearly, the H* adsorption on the strained crystalline region is more favourable than that on amorphous structures, suggesting that the crystalline domain in AC-Ir NSs are the active adsorption sites for HER catalysis instead of amorphous domain, which is also supported by the enhanced HER performance in the interfaced system (Figure R18). Moreover, electrochemical impedance spectroscopy (Figure R19) reveals A-Ir NSs have larger charge-transfer resistance than C-Ir NSs and AC-Ir NSs, revealing A-Ir NSs are probably not the active region for HER catalysis.

Taken together, the existence of amorphous domain contributes to the formation of interface in AC-Ir NSs and further induce the suitable tensile strain in crystalline domain for enhancing HER catalysis. Accordingly, we also discussed this issue in the revised manuscript.

Figure R16. RDF analysis of A-Ir NSs, C-Ir NSs and AC-Ir NSs, respectively.

Figure R17. The calculated ΔG_{H^*} on the amorphous structures and crystalline domain with the 4% tensile strain.

Figure R18. Polarization curves of AC-Ir NSs, C-Ir NSs, A-Ir NSs and Pt/C, respectively.

Figure R19. The electrochemical impedance spectroscopy of AC-Ir NSs, C-Ir NSs and A-Ir NSs, respectively.

Comment 2-3: Considering the much higher price of Ir than that of Pt, it seems non-ideal to replace Pt with Ir in acid media for HER. How about the catalytic performance of AC-Ir NS in alkaline electrolyte?

Response: We thank the reviewer for the valuable suggestions and we agree with the referee that the price of Ir is higher than that of Pt. If the intrinsic activity of Ir can be substantially boosted, it is possible to use low-iridium catalysts to lower the total price of catalysts. Of course, the price is not the key target of this manuscript and the key novelty of this work is to demonstrate the planar strain engineering can revolutionarily change Ir from the electrochemically non-active to highly efficient catalyst toward HER catalysis.

Besides Ir nanosheets, the in-plane strain engineering method is also demonstrated to be a general approach to boost the HER performance of Ru and Rh nanosheets. From the perspective of fundamental science, it is meaningful, because it solves the challenges of strain generation in traditional core-shell nanostructures or metal-substrate interfaces, in which the strained active interface is basically embedded or unexposed by the unstrained region, and thus unfavorable for catalysis.

Furthermore, following the suggestion, the alkaline HER performance of AC-Ir NSs was tested in N_2 -saturated 1.0 M KOH solution with a scan rate of 5 mV/s at 1600 rpm under room temperature. As shown in Figure R20, the AC-Ir NSs require an overpotential of 27 mV at the current of 10 mA cm^{-2} in alkaline medium, which was much less than that of A-Ir NSs (148 mV), C-Ir NSs (188 mV) and Pt/C catalysts (34

mV), suggesting in-plane strain induced by amorphous-crystalline phase boundaries can also boost the alkaline HER catalysis of Ir nanosheets. The related descriptions have been added in the revised manuscript (Page 13, line 284.).

Figure R20. Electrochemical HER performance of AC-Ir NSs in alkaline medium. (a) Polarization curves and (b) Tafel plots of AC-Ir NSs, A-Ir NSs, C-Ir NSs and commercial Pt/C catalysts, respectively. (c) Nyquist plots of catalysts at an overpotential of 20 mV (vs. RHE). (d) Chronoamperometry measurement at an overpotential of 28 mV (vs. RHE) for AC-Ir NSs. Note: All the measurements were conducted in N_2 -saturated 1.0 M KOH solution.

Comment 2-4: It should be not stable structure if you consider such big tensile strain in nanosheets.

Response: We thank the referee for the valuable comments. Considering the drawback of GPA towards strained or misoriented systems as mentioned by the first referee, the average strain tensile may be more meaningful. We have reanalyzed the strain distribution of GPA and also employed other strain probing methods such as electron diffraction profile, the average strain is estimated to be around 4% (see the response to the comment 1-3 of the first referee).

To elucidate the stability of the strain during catalysis, we performed TEM, XRD, aberration-corrected HAADF-STEM and electrochemical active surface area of AC-Ir NSs after ADT. As revealed in Figure R15, R21 and R22, the electrochemical active surface area and structures of AC-Ir NSs can be well maintained after ADT tests. Moreover, the strain analysis of the AC-Ir NSs after ADT was also conducted through EDP analysis. As revealed in Figure R23, the position of main peak of AC-Ir NSs after

ADT remains unchanged, suggesting the tensile strain of AC-Ir NSs is electrochemically stable during HER catalysis. We have added these data in the revised supporting information and given corresponding discussion on it.

Figure R15. The electrochemical active surface area of AC-Ir NSs before and after ADT. (a, b) CO stripping voltammetry of AC-Ir NSs before and after ADT. Stripping of a monolayer of CO in first cycle (red), subsequently second cycle after the stripping of CO (black). (c) Estimation of ECSA of AC-Ir NSs before and after ADT.

Figure R21. Characterizations of AC-Ir NSs before and after ADT. TEM images (a) before and (b) after ADT, respectively. Note: Small particles are commercially activated carbon in TEM images. (c) XRD pattern before and after ADT, respectively.

Figure R22. Aberration-corrected HAADF-STEM images of AC-Ir NSs at different area after ADT.

Figure R23. Electron diffraction profile of AC-Ir NSs after ADT. Electron diffraction profile (a) and selected area electron diffraction pattern (b) of AC-Ir NSs after ADT.

Response to the Reviewer 3:

The manuscript presents a novel catalyst more active for HER than Pt, based on straining Ir. This is a theoretically predicted result, and in itself very interesting. The manuscript goes to great length to argue that the electrocatalytic activity is due to the strain effect. The arguments for this seem pretty solid, given the experimental evidence.

Response: We sincerely thank the reviewer for the positive comments on our work. Based on your valuable comments and suggestions, we have carefully revised our manuscript, shown as follows.

Comment 3-1: A significant amount of carbon and oxygen is observed in the sample, and this will also have a dramatic effect on HER activity, possibly even larger than the strain effect. That the observed increased activity does not originate from impurities can, however, not entirely be excluded (Sup figure 4).

Response: We appreciate the valuable comment raised by the referee. We agree the reviewer's comment that the impurities might affect HER activity of catalyst. Firstly, to affirm the element distribution in AC-Ir NSs, high-resolution EELS spectra were performed. As shown in Figure R5. It is clear that O signal was very limited in both amorphous and crystalline domains. The C signal is mainly observed in the amorphous domains.

Figure R5. (h) Aberration-corrected HAADF-STEM image around amorphous-crystalline phase boundary (inset: dark cyan: amorphous domain, dark green: crystalline domain.). (i) EELS spectra of amorphous domain and crystalline domain corresponding to the selected area in (h), respectively.

Secondly, in order to investigate the influence of C dopants in amorphous Ir domain, we performed the HER performance of A-Ir NSs, whose structure is C-doped amorphous Ir structure. From Figure R24a, the HER activity of A-Ir NSs is inferior to AC-Ir NSs, suggesting crystalline-amorphous phase boundary is important for HER

catalysis. Moreover, DFT calculations in Figure R24b confirm A-Ir NSs with carbon doping possess rather high ΔG_{H^*} (1.47 eV) than AC-Ir NSs (-0.06 eV). Meanwhile, introducing tensile strain of 4% can further manipulate the adsorption behaviour of H^* , revealed by Figure R24c. Combined by the experimental HER performance and the DFT calculations, we believe the improved performance is not simply stemmed from the carbon doping. The carbon doping might contribute to stabilize the amorphous structure and the formed amorphous-crystalline interface.

Thirdly, to further explore the influence from impurities oxygen on HER activity, we tried to intentionally introduce oxygen by increasing the heating temperature and found corresponding iridium oxides at high temperature. The formed oxides exhibit quite poor performance, in comparison with the interface system. Meanwhile, we also performed the DFT calculations of the O-doped amorphous Ir structure and O-doped crystalline Ir structure. From Figure R25, both O-doped amorphous Ir structure and O-doped crystalline Ir structure possess high ΔG_{H^*} value compared with AC-Ir NSs, suggesting the influence from oxygen impurities could be very limited.

Figure R24. (a) Polarization curves of AC-Ir NSs and A-Ir NSs. (b) The calculated ΔG_{H^*} on AC-Ir NSs and C-Ir NSs. (c) The calculated ΔG_{H^*} on AC-Ir NSs with tensile strain of 4% and AC-Ir NSs without tensile strain.

Figure R25. The calculated ΔG_{H^*} of the O-doped amorphous Ir structure and O-doped crystalline Ir structure.

Comment 3-2: The DFT calculations need some further explanation in order to be reproducible. It would be important to know what code was used, what exchange-correlation functional, what part of the structures were fixed, which adsorbate structure corresponds to which color line in Sup. fig. 16, what initial structures/adsorption structures were selected/tested and how was minimum energy structures identified, etc.

Response: We thank the referee for the valuable comments. Following the suggestion, Related details of DFT calculations have been described in “Method” section and as follows: *“The DFT calculations were performed using the Vienna ab initio simulation package VASP. Generalized gradient approximation (GGA) in the form of the Perdew–Burke–Ernzerhof (PBE) function was used as the exchange-correlation interactions. PAW-PBE was employed to describe the interaction between valence electrons and ionic core. The kinetic energy cutoff of 450 eV with spin-polarization was utilized to expand the wave function of the valance electronic states. The van der Waals correction was considered here using the empirical correction in Grimme’s scheme. The Monkhorst-Pack k-point meshes of $3\times 3\times 1$ was used for the Brillouin zone sampling of surface. The convergence tolerances of the optimized calculation were defined as follows: 0.02 eV/Å for the maximum force and 1×10^{-4} eV/atom for the SCF calculation. The ΔG_{H^*} is calculated as follow: $\Delta G_{H^*} = E_{(sur+H^*)} - E_{sur} - 1/2E_{H_2} + 0.24$ eV. The $E_{(sur+H^*)}$ is the energy of surfaces with H adsorption. The E_{H_2} and E_{sur} are the energies of individual H_2 molecule and catalytic surface.”*

Following the referee’s suggestion, the related adsorbate structures were provided in Figure R26. Meanwhile, the related adsorbate structures have been added in the revised supporting information as Figure S19.

Figure R26. The adsorbate structures of AC-Ir NSs with different degrees of lattice expansion. (a) 0% tensile strain, (b) 4% tensile strain, (c) 8% tensile strain, (d) 12% tensile strain, (e) 16% tensile strain and (f) 20% tensile strain, respectively.

Comment 3-3: The manuscript is otherwise excellent, and well worth reading/publishing.

Response: We thank the reviewer very much for the positive comments and the recommendation for publishing.

REVIEWER COMMENTS

Reviewer #1 (Remarks to the Author):

I am very impressed by the thoughtful and extensive additions to this revised manuscript, which engaged with and decisively answered my questions. While the true structure and purity of these complex materials remain unresolved, the authors have assembled a balanced and cohesive narrative which support their conclusions.

If the other reviewer's questions regarding the DFT modelling are resolved, I recommend publication without further revision.

Reviewer #4 (Remarks to the Author):

The manuscript introduces a new method for boosting HER activity of Ir nanosheets by creating in-plane tensile strain arising from a high density of amorphous-crystalline boundaries. It is shown that the design principle also works for other noble metals.

The experimental part is quite impressive introducing a novel method of designing a strain-engineered electrocatalyst with in-plane strain, demonstrating its outstanding performance for HER, and providing an extensive analysis with several very high-quality material characterization techniques supporting the findings. In my opinion, this work could merit publication in Nature Communications after appropriate revisions outlined below. The theoretical part is currently not of sufficiently high standard, and needs to be significantly improved before the manuscript can be recommended for publication.

1. The performance of the newly introduced AC-Ir NSs catalyst and its Ru and Rh analogs has been favorably compared to the commercial Pt/C catalyst. Considering that Ir, Ru and Rh are much more expensive than Pt, why did the authors not attempt to apply the same approach to Pt and prepare an AC-Pt NSs catalyst? To me it looks like a logical step which would demonstrate that not only the chemical nature of the metal but specifically the nanosheet shape together with amorphous-crystalline boundaries boost the HER performance.

2. The enhanced HER performance of the novel Ir catalyst is suggested to be linked to the shifted d-band center as a result of tensile strain. However, the discussion of this aspect is very brief and contradictory. According to the d-band model of Hammer and Nørskov, tensile strain should lead to the upshifting of the d-band center toward the Fermi level and to a stronger interaction between the metal surface and the adsorbates. This is at odds with Fig. 4c, where we can infer that the d-band center of AC-Ir NSs (-2.16 eV) is shifted down relative to C-Ir NSs (-1.95 eV). The description in the manuscript on p. 10 (top) says that the d-band center is shifted up, whereas based on the values of Fig. 4c it is shifted down. It is also stated on p. 10 that "AC-Ir NSs probably possess stronger Ir-H interaction and better electrocatalytic HER performance in acidic media as relative to C-Ir NSs", whereas according to the "Volcano" model of Nørskov et al. (Nørskov et al. J. Electrochem. Soc. 2005, 152, J23-J26), it is the weakest negative H adsorption free energy ($\Delta G_{H^*} \sim 0$) that is optimal for HER performance. Fig. 4c shows that H adsorption free energy is closest to 0 on AC-Ir NSs, as predicted by the computations. Hence, the weak H adsorption on AC-Ir NSs seems to be in agreement with the excellent HER performance but it does not seem to be caused by tensile strain, which would make H adsorption stronger, not weaker compared to C-Ir NSs, and hence, not optimal. Doesn't this mean that not only strain is important but probably other material aspects that distinguish the two types of materials?

3. The section on DFT computations requires significant revisions and additions.

3.1. Currently, a description of chosen models to represent the three materials (C-Ir NSs, A-Ir NSs, and AC-Ir NSs) is completely missing. I can infer that periodic slab models have been used because

periodic code VASP is mentioned. However, it is imperative to explain what exactly has been done, to allow for reproducibility by another researcher after reading this paper. Due to space limitations, these details could be given in the Supplementary information. In particular, I am looking for answers to the following questions:

3.1.1. Was the lattice constant of bulk Ir optimized? If so, what was the optimized Ir-Ir distance and how does it compare to the experimental Ir-Ir distance? If the bulk structure has not been optimized but experimental lattice constant adopted, I would expect some inherent compressive strain introduced with that because the optimal PBE lattice constant would be larger than the experimental one. In that case, stretching by 4% would perhaps relax the structure making it closer to an energy minimum instead of introducing tensile strain.

3.1.2. How were slab models constructed? Which surface (crystallographic direction) was chosen and why? How large was the unit cell in both lateral and vertical dimensions?

3.1.3. How were amorphous and crystalline-amorphous models defined? Were they derived from a crystalline slab by shifting atomic positions? How exactly? I see carbon atoms present in the amorphous regions. At which positions were they placed? How many of them and based on what principle were they distributed? Was the size and shape of the supercells for amorphous and crystalline-amorphous models the same as for the crystalline model?

3.1.4. Were the slab models optimized? If so, which atoms were fixed and which were allowed to change their positions? Were the same atoms allowed to relax when H adsorption was studied?

3.1.5. When tensile strain was applied, was it applied in both x and y directions or only in one direction?

3.1.6. When the adsorption of H atom was studied, at which positions was H atom placed? A figure like Suppl. Fig. 19 is not sufficient because it shows only side views and it does not allow one to understand at which surface site the adsorbed H atom is located. Such a figure should be accompanied by a text describing in more detail at which surface sites the H atom was initially placed and why, whether it remained there after the geometry optimization, and explaining that the geometries and adsorption energies given in Suppl. Fig. 19 correspond to the energy diagrams of Suppl. Fig. 18.

3.2. The "hydrogen adsorption Gibbs free energy (ΔG_{H^*})" is defined by an equation $\Delta G_{H^*} = E(\text{sur}+H^*) - E_{\text{sur}} - 1/2E_{H_2} + 0.24 \text{ eV}$, containing a numerical term 0.24 eV. There is no explanation and no justification given to this approximation and it is not mentioned where it originates from. I think the authors refer to an approximate method introduced by Nørskov group (Nørskov et al. J. Electrochem. Soc. 2005, 152, J23-J26). A reference to the original literature should be given and the applicability of this approximation to the current case should be briefly discussed. Note that the Gibbs free energy depends on temperature but I don't find any temperature dependence in the given definition of ΔG_{H^*} .

3.3. The results shown in Fig. 4 are not sufficiently discussed neither in the main text, nor in the Supplementary information.

3.3.1. In particular, panel (a) shows density difference maps. As it is clear from the term itself, these maps should show a change that occurs when a molecule or a solid changes its state from A to B, for example due to adsorption or another perturbation. These states A and B need to be specified for the whole analysis to be meaningful. Currently they have not been specified. Do the density difference plots refer to density change upon deformation?

3.3.2. Panel (b) shows projected metal d densities of states of the three materials. Do they correspond

to unstretched or stretched systems? Some numerical values are given for each material. I assume that the values refer to the calculated d-band center, but this is not anywhere specified. The method for calculating the d-band center is not specified either. The d-band center position of AC-Ir is intermediate between A-Ir and C-Ir. Is this the expected result? How does it correlate with the HER performance? A discussion is necessary. The specified Monkhorst-Pack k-point mesh of $3 \times 3 \times 1$ is insufficient for accurate DOS plots.

3.3.3. Panel (c) gives the energy diagram which is again not explained. What is the reaction coordinate and what is the middle point on it? I am surprised by a very high positive value of the hydrogen adsorption Gibbs free energy for A-Ir-NSs (1.47 eV) compared to the other three materials, whereas the d-band center position of A-Ir-NSs (-2.22 eV) is closer to AC-Ir-NSs (-2.16 eV) than to C-Ir-NSs (-1.95 eV). This puts in question any correlation between the d-band center position and H adsorption free energy. I guess the energy diagrams and also the DOS plots correspond to stretched AC-Ir-NSs (by 4%) and unstretched other materials but this is not mentioned in the figure caption.

3.3.4. Panel (d). Describe in the caption (or in the legend) which graph corresponds to which vertical axis. I see that the two vertical axes have the corresponding color coding but this is insufficient. Better draw little arrows from a graph toward the corresponding axis. The axis on the left should be called "d-band center" and not just "band center".

3.4. The adsorption free energies of an H atom adsorbed at different positions are shown in the diagrams of Suppl. Fig. 18 (this should be explained in the figure caption) and then one of the values is chosen for each degree of strain and plotted in Suppl. Fig. 20. It is not anywhere explained how the "best" value is chosen. I noticed that in most cases the value with the weakest negative adsorption energy has been chosen. In several places in the manuscript, it is mentioned that DFT predicts "suitable" H* adsorption, but what exactly does "suitable" mean in this context? With $\Delta G_{H^*} \sim 0$?

3.4.1. Suppl. Fig. 20 gives a surprisingly high (positive) adsorption energy for 12% stretched material. From Suppl. Fig. 19, I would conclude that the adsorption energy should be -0.36 eV and not 1.20 eV (I think it is a mistake, and the authors meant 0.12 rather than 1.20).

3.4.2. It would be useful to have an analogous figure to Suppl. Fig. 19 for the H adsorption on C-Ir-NSs and A-Ir-NSs.

3.5. Overall, the shown atomic structures of various surface models are not very helpful because shown is either a top or a side projection of a 3D structure without indicating the depth of various atomic rows, and without any accompanying verbal explanations.

3.6. References to VASP, PBE and PAW should be given. PAW should be spelled out. What was the Grimme's correction scheme, D2 or D3? A reference is required.

Point-by-point response to the referees' comments

We sincerely thank the referees for their careful review and valuable comments, which certainly help improve the manuscript. Our point-by-point responses are presented below and all the changes in the revised manuscript have been highlighted in red for your review.

Response to the Reviewer 1:

I am very impressed by the thoughtful and extensive additions to this revised manuscript, which engaged with and decisively answered my questions. While the true structure and purity of these complex materials remain unresolved, the authors have assembled a balanced and cohesive narrative which support their conclusions. If the other reviewer's questions regarding the DFT modelling are resolved, I recommend publication without further revision.

Response: We thank the reviewer very much for the positive comments and the recommendation for publishing.

Response to the Reviewer 4:

The manuscript introduces a new method for boosting HER activity of Ir nanosheets by creating in-plane tensile strain arising from a high density of amorphous-crystalline boundaries. It is shown that the design principle also works for other noble metals.

The experimental part is quite impressive introducing a novel method of designing a strain-engineered electrocatalyst with in-plane strain, demonstrating its outstanding performance for HER, and providing an extensive analysis with several very high-quality material characterization techniques supporting the findings. In my opinion, this work could merit publication in Nature Communications after appropriate revisions outlined below. The theoretical part is currently not of sufficiently high standard, and needs to be significantly improved before the manuscript can be recommended for publication.

Response: We sincerely appreciate the positive comments on our manuscript. Based on your valuable comments and suggestions, the related details and discussions of slab models have been added in the revised "DFT" section to improve the manuscript. Specific responses are listed below.

1. The performance of the newly introduced AC-Ir NSs catalyst and its Ru and Rh analogs has been favorably compared to the commercial Pt/C catalyst. Considering that Ir, Ru and Rh are much more expensive than Pt, why did the authors not attempt to apply the same approach to Pt and prepare an AC-Pt NSs catalyst? To me it looks like a logical step which would demonstrate that not only the chemical nature of the metal but specifically the nanosheet shape together with amorphous-crystalline boundaries boost the HER performance.

Response: We thank the reviewer for the valuable suggestions. We agree with the referee that the price of Ir/Rh is higher than that of Pt, while Ru is much cheaper than Pt. We also attempt to synthesize the amorphous-crystalline Pt nanosheets (AC-Pt NSs). Unfortunately, neither amorphous Pt NSs nor AC-Pt NSs could not be obtained, might because of strong and isotropic nature of Pt-Pt bonds (*Adv. Mater.* 2018, **30**, 1803234).

2. The enhanced HER performance of the novel Ir catalyst is suggested to be linked to the shifted d-band center as a result of tensile strain. However, the discussion of this aspect is very brief and contradictory. According to the d-band model of Hammer and Nørskov, tensile strain should lead to the upshifting of the d-band center toward the Fermi level and to a stronger interaction between the metal surface and the adsorbates. This is at odds with Fig. 4c, where we can infer that the d-band center of AC-Ir NSs (-2.16 eV) is shifted down relative to C-Ir NSs (-1.95 eV). The description in the manuscript on p. 10 (top) says that the d-band center is shifted up, whereas based on the values of Fig. 4c it is shifted down. It is also stated on p. 10 that “AC-Ir NSs probably possess stronger Ir-H interaction and better electrocatalytic HER performance in acidic media as relative to C-Ir NSs”, whereas according to the “Volcano” model of Nørskov et al. (Nørskov et al. *J. Electrochem. Soc.* 2005, 152, J23-J26), it is the weakest negative H* adsorption free energy ($\Delta G_{H^*} \sim 0$) that is optimal for HER performance. Fig. 4c shows that H* adsorption free energy is closest to 0 on AC-Ir NSs, as predicted by the computations. Hence, the weak H* adsorption on AC-Ir NSs seems to be in agreement with the excellent HER performance but it does not seem to be caused by tensile strain, which would make H* adsorption stronger, not weaker compared to C-Ir NSs, and hence, not optimal. Doesn't this mean that not only strain is important but probably other material aspects that distinguish the two types of materials?

Response: We thank the referee for carefully reviewing our manuscript. We are sorry

for making misdescription of d -band in the original manuscript and have revised it in the revised manuscript: “Fig. 4b reveals the $d_{xz+yz+z^2}$ band center of Ir on AC-Ir surface with 4% tensile strain (-2.16 eV) is located between the values of C-Ir NSs (-1.95 eV) and A-Ir NSs (-2.42 eV), which coincides well with the synchrotron-based valence band spectrum results.” (Page 10 Line 215).

Figure R1. The $d_{xz+yz+z^2}$ band centers of the active Ir sites for AC-Ir NSs with 4% tensile strain, C-Ir NSs and A-Ir NSs, respectively. Note: ϵ_d refers to $E_{d_{xz+yz+z^2}}$.

Firstly, according to the d-band model, the tensile strain can lead to the upshifting of the d-band center toward the Fermi level, which is based on the premise of charge conservation, such as **pure metal system**. However, since carbon atoms are introduced in AC-Ir NSs and A-Ir NSs, the orbital coupling between Ir and C at low energy level can induce the d-band downshift relative to C-Ir NSs.

Furthermore, to further explore the influence from C dopants on HER activity, the DFT calculations of C-doped crystalline Ir NSs was performed. As shown in Figure R2, the C-doped crystalline Ir NSs possess inferior HER performance with rather negative ΔG_{H^*} (-0.29 eV) than AC-Ir NSs (-0.06 eV), which could confirm the C doping cannot be the main reason for performance improvement.

Figure R2. The calculated ΔG_{H^*} of the crystalline Ir with C doping.

Third, we construct the AC-Ir model and the calculation results show the existence of amorphous-crystalline interface can improve the HER performance with more suitable H^* adsorption ($\Delta G_{H^*} = -0.09$ eV). But during the actual synthesis process, the formation of AC-Ir NSs is accompanied by the existence of tensile strain. And Figure R3 shows the intrinsic activity of active sites was further improved ($\Delta G_{H^*} = -0.06$ eV)

and the number of active sites increased in the strained AC-Ir NSs, especially with 4% tensile strain.

Figure R3. The structures of H* adsorption on AC-Ir NSs with different degrees of lattice expansion which are corresponding to the energy diagrams of Figure R5. Note: Here we mainly consider the H* adsorption on Ir top sites of surface four rows near the amorphous-crystalline interface. The unchanged structures with H* adsorption before and after optimization were provided into one graph. Orange, grey and light red spheres in the slab models represent iridium, carbon and hydrogen atoms, respectively.

Besides, the d -band center of the active Ir sites for AC-Ir NSs with different degrees of lattice expansion was calculated (Figure R4), the d -band center has roughly upward trend as the tensile strain increase in AC-Ir NSs. This result implies the strain tensile could affect the position of the $d_{xz+yz+z^2}$ band center, thus affecting the HER catalytic activity.

Figure R4. The $d_{xz+yz+z^2}$ band centers of the active Ir sites for AC-Ir NSs with different degrees of lattice expansion.

Therefore, we believe the tensile strain plays important role on the excellent HER performance of AC-Ir NSs. We have added a brief discussion on the possible other factors which affect the catalytic performance in the revised manuscript.

3. The section on DFT computations requires significant revisions and additions.

Response: Following the suggestions, we have performed significant revision and additions, as discussed below.

3.1. Currently, a description of chosen models to represent the three materials (C-Ir NSs, A-Ir NSs, and AC-Ir NSs) is completely missing. I can infer that periodic slab models have been used because periodic code VASP is mentioned. However, it is imperative to explain what exactly has been done, to allow for reproducibility by another researcher after reading this paper. Due to space limitations, these details could be given in the Supplementary information. In particular, I am looking for answers to the following questions:

Response: Thanks for your valuable suggestion. Here we add more description about the construction of all the models. The detail information was discussed as below and added in the revised manuscript and supporting information.

3.1.1. Was the lattice constant of bulk Ir optimized? If so, what was the optimized Ir-Ir distance and how does it compare to the experimental Ir-Ir distance? If the bulk structure has not been optimized but experimental lattice constant adopted, I would expect some inherent compressive strain introduced with that because the optimal PBE lattice constant would be larger than the experimental one. In that case, stretching by

4% would perhaps relax the structure making it closer to an energy minimum instead of introducing tensile strain.

Response: We thank the reviewer for the valuable suggestions. Firstly, we have optimized the lattice constant of bulk Ir and structure comes convergence in one ion step. The change of lattice constant is less than 2%, which proves the applicability of the parameters in this system. The Ir-Ir bond is 2.715 Å which is same to the experimental Ir-Ir distance. Under the calculation parameters, the Ir-Ir distance and the lattice constant of Ir does not change compared with the experimental Ir-Ir distance, which eliminates the possibility of strain caused by self-optimization. In addition, the stretching from 0-20% are applied to study the effect of tensile strain on the HER performance.

3.1.2. How were slab models constructed? Which surface (crystallographic direction) was chosen and why? How large was the unit cell in both lateral and vertical dimensions?

Response: We thank the referee for carefully reviewing our manuscript. The slab model of crystalline structure was constructed by cleaving (0 -1 1) plane of Ir bulk. For the (0 -1 1) plane is perpendicular to the (2 0 0) and (1 1 1) planes, which coincides well with the FFT patterns of crystalline structure in Figure 1. And (0 -1 1) plane is considered as the crystal plane in observation direction. The models of A-Ir and AC-Ir are built by doping C at the site of Ir vacancy. The position of C atoms is determined by the EXAFS results and the coordination numbers of model structures for AC-Ir NSs and A-Ir NSs are close with the results of XAFS analysis. Besides we apply the tensile strain in the x direction with different ratio for AC-Ir. The surface is modeled by a periodic six-layer slab repeated in 3*2 surface unit cell with a vacuum region of 12 Å between the slabs along the Z axis.

Following the referee's suggestion, the related details of model construction have been added in the revised "DFT" section.

3.1.3. How were amorphous and crystalline-amorphous models defined? Were they derived from a crystalline slab by shifting atomic positions? How exactly? I see carbon atoms present in the amorphous regions. At which positions were they placed? How many of them and based on what principle were they distributed? Was the size and shape of the supercells for amorphous and crystalline-amorphous models the same as for the crystalline model?

Response: We thank the referee for the constructive comments. Firstly, we constructed models of A-Ir NSs and AC-Ir NSs in our work based on the coordination structural information obtained from XAFS analysis (Table R1). Specifically, the models of A-Ir and AC-Ir are built by doping C at the site of Ir vacancy. The position of C atoms is determined by the EXAFS results and the coordination numbers of model structures for AC-Ir NSs and A-Ir NSs are close with the results of XAFS analysis (Table R2). The size and shape of the supercells for A-Ir NSs and AC-Ir NSs are a periodic six-layer slab repeated in 3*2 surface unit cell with a vacuum region of 12 Å between the slabs along the Z axis, which is the same as the C-Ir NSs model. The models of AC-Ir NSs with different degrees tensile stain are similar, except that we apply the tensile strain in the x direction with different ratio (Figure R5).

Table R1. EXAFS fitting parameters at the Ir L₃-edge for Ir powder, A-Ir NSs and AC-Ir NS (S₀²=0.807).

Sample	Shell	N ^a	R(Å) ^b	σ ² (Å ²) ^c	ΔE ₀ (eV) ^d	R factor
Ir powder	Ir-Ir	12.0	2.71	0.0031	8.3	0.0012
A-Ir NSs	Ir-C	5.1	2.01	0.0018	9.4	0.0013
	Ir-Ir	6.3	2.92	0.0076		
AC-Ir NSs	Ir-C	4.5	2.00	0.0050	9.3	0.0065
	Ir-Ir	7.2	2.90	0.0076		

Note: ^aN: coordination numbers; ^bR: bond distance; ^cσ²: Debye-Waller factors; ^dΔE₀: the inner potential correction.

Table R2. The coordination numbers of AC-Ir NSs, A-Ir NSs and C-Ir NSs based on model structures.

Theoretical model	Shell	Coordination numbers
C-Ir NSs	Ir-Ir	10.0
A-Ir NSs	Ir-C	4.1
	Ir-Ir	5.8
AC-Ir NSs	Ir-C	4.0
	Ir-Ir	7.3

Figure R5. The slab models of C-Ir, A-Ir and AC-Ir before and after optimization. Note: The size and shape of slab model are a periodic six-layer slab repeated in 3*2 surface unit cell with a vacuum region of 12 Å between the slabs along the Z axis. Orange and grey spheres in the slab models represent iridium and carbon atoms, respectively.

3.1.4. Were the slab models optimized? If so, which atoms were fixed and which were allowed to change their positions? Were the same atoms allowed to relax when H adsorption was studied?

Response: We thank the referee for the valuable comments. The slab model is optimized with tensile strain in the x direction. Here all the atoms are relaxed for optimization and the lattice is fixed. When H* adsorption was studied, the same atoms were allowed to relax.

3.1.5. When tensile strain was applied, was it applied in both x and y directions or only in one direction?

Response: We are pleased to clarify this issue. Because the x direction in slab model is perpendicular to the amorphous-crystalline interface, which is consistent with the strain analysis, we apply the tensile strain in the x direction with different ratio to study the strain effect. Furthermore, the related details of slab models have been added in the revised “DFT” section.

3.1.6. When the adsorption of H atom was studied, at which positions was H atom placed? A figure like Suppl. Fig. 19 is not sufficient because it shows only side views and it does not allow one to understand at which surface site the adsorbed H atom is

located. Such a figure should be accompanied by a text describing in more detail at which surface sites the H atom was initially placed and why, whether it remained there after the geometry optimization, and explaining that the geometries and adsorption energies given in Suppl. Fig. 19 correspond to the energy diagrams of Suppl. Fig. 18.

Response: We thank the referee for carefully reviewing our manuscript. In order to clearly show the sites of H* adsorption, we rotated the display angle as shown in Figure R2. Here we mainly consider the H* adsorption on Ir top sites of surface four rows near the amorphous-crystalline interface. Furthermore, the unchanged structures with hydrogen adsorption before and after optimization were provided into one graph. The others are divided into before optimized and after optimized structures to see the changes of adsorption sites.

Furthermore, following the referee's suggestion, the detailed description have been added in the figure caption of Suppl. Fig. 18 and Suppl. Fig. 19 in the revised supporting information.

Figure R2. The structures of H* adsorption on AC-Ir NSs with different degrees of lattice expansion which are corresponding to the energy diagrams of Figure R5. Note: Here we mainly consider the H* adsorption on Ir top sites of surface four rows near the amorphous-crystalline interface. The unchanged structures with H* adsorption before and after optimization were provided into one graph. Orange, grey and light red spheres in the slab models represent iridium, carbon and hydrogen atoms, respectively.

Figure R6. The calculated ΔG_{H^*} on different Ir active sites of the strained AC-Ir NSs: (a) 0% tensile strain, (b) 4% tensile strain, (c) 8% tensile strain, (d) 12% tensile strain, (e) 16% tensile strain and (f) 20% tensile strain, respectively.

3.2. The “hydrogen adsorption Gibbs free energy (ΔG_{H^*})” is defined by an equation $\Delta G_{H^*} = E_{(sur+H^*)} - E_{sur} - 1/2E_{H_2} + 0.24$ eV, containing a numerical term 0.24 eV. There is no explanation and no justification given to this approximation and it is not mentioned where it originates from. I think the authors refer to an approximate method introduced by Nørskov group (Nørskov et al. J. Electrochem. Soc. 2005, 152, J23-J26). A reference to the original literature should be given and the applicability of this approximation to the current case should be briefly discussed. Note that the Gibbs free energy depends on temperature but I don’t find any temperature dependence in the given definition of ΔG_{H^*} .

Response: We appreciate the valuable comment raised by the referee. Following the suggestion, we have cited the references in the revised manuscript as references 40. Here the $\Delta G_{H^*} = \Delta E_H + \Delta E_{ZPE} - T\Delta S_H$. $\Delta S_H \approx -1/2\Delta S_{H_2}^0$, where $S_{H_2}^0$ is the entropy of H_2 in the gas phase at standard conditions. ΔE_{ZPE} is the difference in zero point energy between the adsorbed and the gas phase. Furthermore, the calculated temperature is 300 K which is close to the room temperature. All these means that $\Delta G_{H^*} = \Delta E_H + 0.24$ eV. Meanwhile, the related details of DFT calculations have been

described in “Method” section.

3.3. The results shown in Fig. 4 are not sufficiently discussed neither in the main text, nor in the Supplementary information.

Response: We thank the referee for the constructive comments. Follow your suggestion, we have sufficiently discussed the results shown in Fig. 4 in the revised manuscript and Supplementary information. The main details are as follows.

3.3.1. In particular, panel (a) shows density difference maps. As it is clear from the term itself, these maps should show a change that occurs when a molecule or a solid changes its state from A to B, for example due to adsorption or another perturbation. These states A and B need to be specified for the whole analysis to be meaningful. Currently they have not been specified. Do the density difference plots refer to density change upon deformation?

Response: We thank the referee for the valuable comments and we have specified it in the revised manuscript. The density difference is the deformation charge density following the formula $\rho = \rho_{\text{sur}} - \sum_{i=1}^m \rho_{\text{Ir}_m} - \sum_{i=1}^n \rho_{\text{C}_n}$. Here ρ_{sur} is the charge density of surface. $\sum_{i=1}^m \rho_{\text{Ir}_m}$ and $\sum_{i=1}^n \rho_{\text{C}_n}$ are the sum of charge density of free iridium atom and carbon atom, respectively. It is calculated by subtracting the charge density of single atom from the charge density of the whole surface. In order for intuitive comparison, only the charge accumulation is shown. We change to deformation charge density in the figure caption of Fig. 4a. Meanwhile, the related details of the deformation charge density have been added in the revised “DFT” Method.

3.3.2. Panel (b) shows projected metal d densities of states of the three materials. Do they correspond to unstretched or stretched systems? Some numerical values are given for each material. I assume that the values refer to the calculated d-band center, but this is not anywhere specified. The method for calculating the d-band center is not specified either. The d-band center position of AC-Ir is intermediate between A-Ir and C-Ir. Is this the expected result? How does it correlate with the HER performance? A discussion is necessary. The specified Monkhorst-Pack k-point mesh of $3 \times 3 \times 1$ is insufficient for accurate DOS plots.

Response: We thank the referee for the constructive comments. The AC-Ir NSs in Panel (b) is correspond to the 4% stretched surface. The numerical values are the band

center of $d_{xz+yz+z^2}$. To specify the value of the calculated d -band center, we have revised Figure 4b and added the label of $E_{d_{xz+yz+z^2}}$ in the figure caption (Figure R1). It is calculated by the formula $E = \frac{\int_{-\infty}^{\infty} yx dx}{\int_{-\infty}^{\infty} y dx}$, y is the density of states, x is the energy. The metal Ir and H interaction will produce bonding state and antibonding state. The antibonding state near the Fermi level (E_F) mainly decide the bonding strength of Ir-H. The downshift of d band will result antibonding state downshift and the more electron filling of anti-bonding orbital contribute to the weaker Ir-H interaction.

Furthermore, following the referee's suggestion, we recalculate PDOS with Monkhorst-Pack k-point mesh of $6 \times 6 \times 1$ as shown in Figure R7. The $d_{xz+yz+z^2}$ band centers show small change compare to the results calculated by Monkhorst-Pack k-point mesh of $3 \times 3 \times 1$. The $d_{xz+yz+z^2}$ band center downshifts in the strained AC-Ir NSs compare to C-Ir NSs. The ΔG_{H^*} reaches the optimal value at around 4% tensile strain. These confirm the tensile strain plays important role on the excellent HER performance of AC-Ir NSs.

Meanwhile, we have specified it and made detailed discussion about $d_{xz+yz+z^2}$ band centers in the manuscript: *“Additionally, the $d_{xz+yz+z^2}$ band center of the Ir sites and the ΔG_{H^*} of AC-Ir NSs as a function of tensile strain are summarized in Fig. 4d and Supplementary Fig. 18-21. It shows a rough trend in upward shift of the band centers as the tensile strain increase in AC-Ir NSs, which always downshifts compared to C-Ir NSs. Meanwhile, the ΔG_{H^*} reaches the optimal value at around 4% tensile strain among different degrees of lattice expansion, revealing the 4% tensile strain plays an important role on enhancing the intrinsic HER activity.”*

Figure R7. The $d_{xz+yz+z^2}$ band centers of the active Ir sites for C-Ir NSs, A-Ir NSs and AC-Ir NSs with different degrees of lattice expansion. Note: ϵ_d refers to $E_{d_{xz+yz+z^2}}$.

3.3.3. Panel (c) gives the energy diagram which is again not explained. What is the reaction coordinate and what is the middle point on it? I am surprised by a very high positive value of the hydrogen adsorption Gibbs free energy for A-Ir-NSs (1.47 eV) compared to the other three materials, whereas the d-band center position of A-Ir-NSs (-2.22 eV) is closer to AC-Ir-NSs (-2.16 eV) than to C-Ir-NSs (-1.95 eV). This puts in question any correlation between the d-band center position and H adsorption free energy. I guess the energy diagrams and also the DOS plots correspond to stretched AC-Ir-NSs (by 4%) and unstretched other materials but this is not mentioned in the figure caption.

Response: We thank the referee for the valuable comments. The HER process can be described using the following equation: $H^+ + e^- \rightleftharpoons H^* \rightleftharpoons 1/2H_2$. At $T=300K$, $P_{H_2} = 1$ bar and $pH = 0$, the ΔG for this equation is zero. The middle point is H^* .

Due to the disorder of amorphous structure, there are so many surface sites. Here we further calculate other sites. The result shows the optimized ΔG_{H^*} of A-Ir-NSs is 0.22

eV. In addition, the position of d -band can only predict the trend of adsorption energy, and it is difficult to fit the two values completely linearly. Relatively speaking, ΔG_{H^*} can reflect the catalytic performance more directly.

Furthermore, the energy diagrams and the PDOS plots correspond to stretched AC-Ir-NSs (by 4%) and unstretched other materials. To make it clear for readers, we have specified it in the figure captions in the revised the manuscript.

3.3.4. Panel (d). Describe in the caption (or in the legend) which graph corresponds to which vertical axis. I see that the two vertical axes have the corresponding color coding but this is insufficient. Better draw little arrows from a graph toward the corresponding axis. The axis on the left should be called “d-band center” and not just “band center”.

Response: We thank the reviewer for the valuable suggestions. Following the referee’s suggestions, we add little arrows and specify $d_{xz+yz+z^2}$ band center in Figure R8 and the revised Figure 4d.

Figure R8. The $d_{xz+yz+z^2}$ band centers of the active Ir sites and the calculated best ΔG_{H^*} on AC-Ir NSs with different degrees of lattice expansion.

3.4. The adsorption free energies of an H atom adsorbed at different positions are shown in the diagrams of Suppl. Fig. 18 (this should be explained in the figure caption) and then one of the values is chosen for each degree of strain and plotted in Suppl. Fig. 20. It is not anywhere explained how the “best” value is chosen. I noticed that in most cases the value with the weakest negative adsorption energy has been chosen. In several places in the manuscript, it is mentioned that DFT predicts “suitable” H^* adsorption, but what exactly does “suitable” mean in this context? With $\Delta G_{H^*} \sim 0$?

Response: Thank you for the suggestion. Following the referee’s suggestion, the related explanation have been added in the figure captions of Suppl. Fig. 18 and Suppl.

Fig. 20 in revised supporting information. In DFT calculation of HER catalysis, the ΔG_{H^*} close to neutral is considered as the suitable H^* adsorption, so the minimum value is considered as the best ΔG_{H^*} for each degree of strain. To avoid misleading, we have explained the “suitable H^* adsorption” in the revised manuscript.

3.4.1. Suppl. Fig. 20 gives a surprisingly high (positive) adsorption energy for 12% stretched material. From Suppl. Fig. 19, I would conclude that the adsorption energy should be -0.36 eV and not 1.20 eV (I think it is a mistake, and the authors meant 0.12 rather than 1.20).

Response: We sincerely thank the referee for carefully reviewing our manuscript and sincerely apologize for the spelling errors. It is 0.12 eV instead of 1.2 eV, which has been corrected in Figure R9. The ΔG_{H^*} close to neutral is considered as the suitable H^* adsorption. The value of -0.36 eV suggests stronger adsorption which could hinder the desorption of H_2 on surface compared with 0.12 eV.

Figure R9. The best ΔG_{H^*} comparison between interfacial models with different degrees of lattice expansion.

3.4.2. It would be useful to have an analogous figure to Suppl. Fig. 19 for the H adsorption on C-Ir-NSs and A-Ir-NSs.

Response: We thank the reviewer for the valuable suggestions. Following the suggestions, the related structures of H^* adsorption on A-Ir-NSs and C-Ir-NSs were provided in Figure R10. Meanwhile, the related structures have been added in the revised supporting information as Suppl. Fig. 17.

Figure R10. The structures of H^* adsorption on on A-Ir-NSs and C-Ir-NSs, respectively. Note: The unchanged structures with H^* adsorption before and after optimization were provided into one graph. Orange, grey and light red spheres in the slab models represent iridium, carbon and hydrogen atoms, respectively.

3.5. Overall, the shown atomic structures of various surface models are not very helpful because shown is either a top or a side projection of a 3D structure without indicating the depth of various atomic rows, and without any accompanying verbal explanations.

Response: We appreciate the valuable comment raised by the referee. Following the suggestion, we rotated the display angle to show the depth of various atomic rows and shown in Figure R3 and in the revised supporting information as Suppl. Fig. 18. Meanwhile, the related explanation has been added in the figure caption.

3.6. References to VASP, PBE and PAW should be given. PAW should be spelled out. What was the Grimme's correction scheme, D2 or D3? A reference is required.

Response: We appreciate the valuable comment raised by the referee. Following the referee's suggestion, the related references (Phys. Chem. Chem. Phys., 19, 32184 (2017)) was cited in the revised manuscript as references 39. For the Grimme's correction, we use the zero damping DFT-D3, which has been pointed out in the revised manuscript.

REVIEWER COMMENTS

Reviewer #4 (Remarks to the Author):

The authors improved the Computational Methods section and corrected some of the issues I have raised in my previous report, but some of my points raised previously remained not addressed to my satisfaction. The main problem remains the lack of clear and convincing procedure for comparing the adsorption energies of the H atom on different Ir surfaces. See more detailed comments below.

1. It is still not clear how the models have been constructed, although the authors have provided some more detail compared to the previous version. New Suppl. Fig. 15 is a step in the right direction. However, it remains unclear at which specific positions within the slab C impurities were added and how many impurity atoms per unit cell. It is not sufficient to say that "the position of C atoms is determined by the EXAFS results", this does not help me if I would like to reproduce the calculation or if I simply would like to understand the set of rules used to create the A-Ir and AC-Ir models. The authors need to provide a figure (perhaps combining top and side views of the Ir(0-11) slab) explaining clearly where C atoms were placed in which case. It seems that for the AC-Ir material, C atoms were placed in one plane normal to the surface, whereas to create the amorphous material, they were placed all over the place, though I cannot see if there is any pattern or the substitution was random.

2. The various adsorption positions considered for H adsorption also remain unclear. It is very difficult to understand them from Suppl. Fig. 18 or Fig. R2. Why is the number of considered adsorption sites different for surfaces with different degree of stretching? I request that the authors provide a clear figure with the top view of the surface (could be schematic) showing which adsorption positions have been considered for the three studied slab models. These positions could be labelled and indicated for each geometry shown in Suppl. Fig. 18 to facilitate understanding.

3. In my previous report (comment 3.4) I pointed out that it was not specified how the "best" value of the adsorption free energy was chosen. Now the authors gave an explanation, but only in the caption of Supplementary Figure 20, where it may be difficult to find, while this is an important concept in the whole computational analysis. It is stated that "the minimum value is considered as the best ΔGH^* for each degree of strain". But is it logical to compare the minimum value of the adsorption energy? On every metal surface, one could find certain less favorable adsorption sites, but these sites are not so important for reactivity because adsorption of H would occur on strongly binding sites. Therefore, in my opinion, one should be comparing the adsorption energies of most strongly binding sites, rather than least strongly binding ones.

4. Previously, I commented that an approximate formula for computing "the hydrogen adsorption free Gibbs free energy" was given but it was not explained. Now, the authors provided a reference to the article where the formula originates from, but I feel that a brief comment is needed on what the numerical term 0.24 eV stands for. Furthermore, for the sake of rigor, this property should be called "the standard Gibbs free energy", since it refers to standard conditions. Clearly, this is a very approximate formula, because the difference between the adsorption energy (at 0K in vacuum) and ΔG is given by a constant term, same for all adsorption sites. But this is acceptable because the analysis is qualitative.

5. In their response to my comment (2) of the previous report the authors compared H adsorption on C-doped crystalline and amorphous Ir and concluded that "C-doped crystalline Ir NSs possess inferior HER performance with rather negative ΔGH^* (-0.29 eV) than AC-Ir NSs (-0.06 eV)". Further, they concluded that "C doping cannot be the main reason for performance improvement". I am not convinced by this analysis because the adsorption sites that have been compared seem to be different: on-top site for AC-Ir and bridge site for CC-Ir. Also, the number and location of C impurities

and the proximity of the adsorption site to C impurities would strongly influence the adsorption energy. There are many different adsorption sites on both surfaces with different adsorption energies and hence all of them need to be considered, otherwise a comparison is not valid.

Figure R3 compares the H adsorption energies on surfaces with different degree of stretching. The authors argue that the value of H adsorption energy on a surface stretched by 4%, -0.06 eV, is slightly improved compared to unstretched surface, -0.09 eV. But in my opinion, the difference of 0.03 eV is so minor that one cannot really call it a difference and make any far going conclusions based on that. Furthermore, it seems that the adsorption sites being compared are again not equivalent. In general, it is hard to see which adsorption sites are equivalent when looking at panels a-f of Figure R3. It seems that the location of C impurities is changing when surfaces are stretched, and even the number of C atoms per unit cell seems to be changing (8 in structures shown in row f versus 6 elsewhere). Hence, I don't think it is fair to compare them. A comparison should be systematic, so that the same surface composition and the same type of adsorption site are being compared against each other. If stretching causes restructuring, that should be commented on.

Figure 4. I don't agree with the interpretation of Figure 4(a) in terms of the transfer of electrons from crystalline domains to amorphous domains in AC-Ir NSs. To me, it rather looks like the electrons are being transferred from Ir to a more electronegative C.

The English is in general good but some sentences suffer from wrong word choice and partially incorrect grammar. For example, I think that the expression "suitable Gibbs free energy" or "suitable H* adsorption" is wrong word choice. Probably, it should be "favorable" or "optimal" Gibbs free energy.

Here is another example. "Generalized gradient approximation (GGA) in the form of the Perdew–Burke–Ernzerhof (PBE) function with DFT-D3 scheme was used as the exchange-correlation interactions. Projector augmented wave (PAW) was employed to describe the interaction between valence electrons and ionic core."

This passage could be improved as follows: "The Perdew–Burke–Ernzerhof (PBE) form of the generalized gradient approximation (GGA) [ref] was used to describe the exchange-correlation functional. Dispersion effects have been accounted for according to Grimme's correction scheme (DFT-D3) [Grimme, S.; Antony, J.; Ehrlich, S. and Krieg, H., J. Chem. Phys., 2010, 132, 154104.] Projector augmented wave (PAW) method was employed to describe the interaction between valence electrons and ionic cores."

"Comprehension on the effect of tensile strain" sounds a bit strange. I would change it to "Understanding the effect of tensile strain".

I recommend proof-reading by a language expert who could improve the sentence structure. Because this is a top-tier journal, I also expect that the quality of written English is of sufficiently high standard.

Point-by-point response to the referees' comments

We sincerely thank the referee for the careful review and valuable comments, which certainly help improve the manuscript. Our point-by-point responses are presented below and all the changes in the revised manuscript have been highlighted in red for your review.

Response to the Reviewer 4:

The authors improved the Computational Methods section and corrected some of the issues I have raised in my previous report, but some of my points raised previously remained not addressed to my satisfaction. The main problem remains the lack of clear and convincing procedure for comparing the adsorption energies of the H atom on different Ir surfaces. See more detailed comments below.

Response: We are sorry for the unsatisfactory revision in the previous version. Based on your valuable comments and suggestions, the related details and discussions of slab models and the schematic sites for the H* adsorption on different surfaces have been added in the revised "Theoretical calculation methods" section. Specific responses are listed below.

1. It is still not clear how the models have been constructed, although the authors have provided some more detail compared to the previous version. New Suppl. Fig. 15 is a step in the right direction. However, it remains unclear at which specific positions within the slab C impurities were added and how many impurity atoms per unit cell. It is not sufficient to say that "the position of C atoms is determined by the EXAFS results", this does not help me if I would like to reproduce the calculation or if I simply would like to understand the set of rules used to create the A-Ir and AC-Ir models. The authors need to provide a figure (perhaps combining top and side views of the Ir(0-11) slab) explaining clearly where C atoms were placed in which case. It seems that for the AC-Ir material, C atoms were placed in one plane normal to the surface, whereas to create the amorphous material, they were placed all over the place, though I cannot see if there is any pattern or the substitution was random.

Response: Following the suggestions, we describe the construction of models in more details. The side and top view of the constructed slab models for A-Ir NSs and AC-Ir NSs are shown in Figure R1. We initially cleave the Ir (0 -1 1) by a periodic six-layer slab repeated in 3×2 surface unit cell and get surface slab with 72 Ir atoms (six layers atoms). To construct A-Ir model, we replace 24 Ir atoms by C and the ratio of C:Ir is 24:48. For the I, III, IV and V layers, Ir and C atoms are arranged one by one. The II and VI layer are all Ir atoms. To construct AC-Ir model, we replace 8 Ir atoms by C and the ratio of C:Ir is 8:64. Ir and C atoms are arranged one by one in the I, III, V layers of the fifth column, and IV layer of the sixth column. All the others are Ir atoms. From the HRTEM image in Figure 2a, we can see the *amorphous-crystalline boundary*, so we simulate the crystalline domain and amorphous domain from left to right along x axis in AC-Ir, and C atoms are listed around boundary. Furthermore, the coordination numbers of optimized model structures for AC-Ir NSs and A-Ir NSs are close with the

results of XAFS analysis. In order to compare the adsorption of intermediates on the catalytic surfaces, the two-dimensional periodic Ir (0 -1 1) and A-Ir models were created, and which are used as control groups.

Figure R1. The side and top views of the constructed slab models for (a) A-Ir and (b) AC-Ir.

2. The various adsorption positions considered for H adsorption also remain unclear. It is very difficult to understand them from Suppl. Fig. 18 or Fig. R2. Why is the number of considered adsorption sites different for surfaces with different degree of stretching? I request that the authors provide a clear figure with the top view of the surface (could be schematic) showing which adsorption positions have been considered for the three studied slab models. These positions could be labelled and indicated for each geometry shown in Suppl. Fig. 18 to facilitate understanding.

Response: We thank the referee for the valuable suggestion. Here the schematic H* adsorption on AC-Ir, C-Ir and A-Ir is shown in Figure R2. Since the interface between column 2 and 3 is considered as the *amorphous-crystalline boundary*, we mainly calculate the H* adsorption on Ir and C top sites from column 1 to 4 as shown in Figure R2a. For the ordered model of AC-Ir before relaxation, the H* adsorption on Ir top sites are same sites in AC-Ir model. Due to the structural relaxation in the process of structural optimization, the optimized structures and atoms of AC-Ir with different tensile strain become disordered. Therefore, we calculate the adsorption energies of H* on all designed Ir and C sites in different columns according to the optimized structures. It should be noted that some of sites have the same atomic environments, and only one of them are calculated, and some of sites have same energies. Overall, , the calculated ΔG_{H^*} are shown in Figure R3.

In Figure R2b, the Ir atoms of column 1 and column 2 are in the surface layer and sublayer, respectively. So we calculate the H* adsorption at the sites of Ir10, Ir44 and Ir10-Ir26. Similarly, we calculate the H* adsorption at the sites of Ir23, Ir14, Ir47 for

A-Ir as shown in Figure R2c.

Figure R2. The schematic sites for H^* adsorption on (a) AC-Ir, (b) C-Ir and (c) A-Ir.

Figure R3. The calculated ΔG_{H^*} on different Ir active sites of the strained AC-Ir NSs with (a) 0%, (b) 4%, (c) 8%, (d) 12%, (e) 16%, (f) 20% tensile strain, respectively. Note: the number is denoted as the location marked in Figure R2a.

3. In my previous report (comment 3.4) I pointed out that it was not specified how the “best” value of the adsorption free energy was chosen. Now the authors gave an explanation, but only in the caption of Supplementary Figure 20, where it may be difficult to find, while this is an important concept in the whole computational analysis. It is stated that “the minimum value is considered as the best ΔG_{H^*} for each degree of strain”. But is it logical to compare the minimum value of the adsorption energy? On every metal surface, one could find certain less favorable adsorption sites, but these sites are not so important for reactivity because adsorption of H would occur on strongly binding sites. Therefore, in my opinion, one should be comparing the adsorption energies of most strongly binding sites, rather than least strongly binding ones.

Response: We appreciate the valuable comments. In the DFT calculation of HER catalysis, since the ΔG_{H^*} close to neutral is extensively considered as the favorable H^* adsorption [Jaramillo et al., Science, 2007, 317, 100-102], the minimum absolute value is considered as the optimal ΔG_{H^*} for each degree of strain (Figure R4). We have added a brief description on it in the revised manuscript.

[redacted]

We recognize that there are sites with strong hydrogen adsorption on different surfaces and hydrogen adsorption occurs at these sites. However, according to the adsorption volcanic curve, the HER catalytic rate decrease with decreasing ΔE_H due to a lack of available sites for $H + H$ recombination at the surface. The strongly binding sites are not conducive to the combination of adsorbed hydrogen to produce hydrogen molecule [Nørskov et al., J. Electrochem. Soc., 2005, 152, J23-J26]. Therefore, we did not consider these strong adsorption sites as the active sites for the optimal catalytic reaction.

4. Previously, I commented that an approximate formula for computing “the hydrogen

adsorption free Gibbs free energy” was given but it was not explained. Now, the authors provided a reference to the article where the formula originates from, but I feel that a brief comment is needed on what the numerical term 0.24 eV stands for. Furthermore, for the sake of rigor, this property should be called “the standard Gibbs free energy”, since it refers to standard conditions. Clearly, this is a very approximate formula, because the difference between the adsorption energy (at 0K in vacuum) and ΔG is given by a constant term, same for all adsorption sites. But this is acceptable because the analysis is qualitative.

Response: Thank you for your valuable suggestion. Following the suggestion, we have modified “the standard Gibbs free energy” in the revised manuscript. ΔE_{ZPE} is the difference in zero point energy between the adsorbed and the gas phase, the value of which is 0.17 and 0.135, respectively. ΔE_{ZPE} is calculated to be 0.04 eV. $\Delta S_H \approx -1/2\Delta S_{H_2}^0$ due to the small value of vibrational entropy in the adsorbed state, where $S_{H_2}^0$ is the entropy of H_2 in the gas phase at standard conditions. At the temperature of 300 K, $T\Delta S$ is calculated to be -0.2 eV. So we use 0.24 eV to represent $\Delta E_{ZPE} - T\Delta S_H$. In the revised manuscript, we have added the details and gave a brief discussion on it.

5. In their response to my comment (2) of the previous report the authors compared H adsorption on C-doped crystalline and amorphous Ir and concluded that “C-doped crystalline Ir NSs possess inferior HER performance with rather negative ΔG_{H^*} (-0.29 eV) than AC-Ir NSs (-0.06 eV)”. Further, they concluded that “C doping cannot be the main reason for performance improvement”. I am not convinced by this analysis because the adsorption sites that have been compared seem to be different: on-top site for AC-Ir and bridge site for CC-Ir. Also, the number and location of C impurities and the proximity of the adsorption site to C impurities would strongly influence the adsorption energy. There are many different adsorption sites on both surfaces with different adsorption energies and hence all of them need to be considered, otherwise a comparison is not valid.

Response: We thank the referee for the valuable comments. We recognize that carbon doping could improve the catalytic hydrogen evolution performance of crystalline Ir. However, it could not be the main reason for the performance improvement, because the optimal ΔG_{H^*} of the AC-Ir with different tensile strain are more favorable than that on C-doped Ir NS, which are respectively -0.09, -0.06, -0.16, 0.12, -0.27, -0.28 eV for 0%, 4%, 8%, 12%, 16%, 20% strained AC-Ir, and -0.29 eV for C-doped crystalline Ir NS.

We agree that the number and location of C impurities would influence the adsorption energy. Here we qualitatively discuss the effect of carbon doping in the crystal Ir on hydrogen adsorption. The surface of C-doped Ir constructed by doping four C atoms in crystalline Ir with 72 Ir atoms. As shown in Figure R5a, the surface of crystalline Ir with C doping become local disordered after optimization. So, we only consider the H adsorption on the Ir sites around the C dopants. It shows that the optimal ΔG_{H^*} is -0.29 eV at Ir-Ir bridge site labelled as 4 (Figure R5a), indicating that the ΔG_{H^*} on C-doped Ir can be improved from -0.37 eV to -0.29 eV after C doping. Although the

C dopants can enhance the HER intrinsic activity, the C-doped Ir possesses the inferior activity compared with AC-Ir with 4% tensile strain (-0.06 eV). The improvement of hydrogen adsorption on AC-Ir with tensile strain is obviously larger than that on C-doped Ir. So, we believe that the C doping cannot be the main reason for the performance improvement.

Figure R5. (a) The side and top views of optimized C-doped Ir surface. (b) The calculated ΔG_{H^*} of C-doped Ir (The number is denoted as the location marked in Figure R5a).

Figure R3 compares the H adsorption energies on surfaces with different degree of stretching. The authors argue that the value of H adsorption energy on a surface stretched by 4%, -0.06 eV, is slightly improved compared to unstretched surface, -0.09 eV. But in my opinion, the difference of 0.03 eV is so minor that one cannot really call it a difference and make any far going conclusions based on that. Furthermore, it seems that the adsorption sites being compared are again not equivalent. In general, it is hard to see which adsorption sites are equivalent when looking at panels a-f of Figure R3. It seems that the location of C impurities is changing when surfaces are stretched, and even the number of C atoms per unit cell seems to be changing (8 in structures shown in row f versus 6 elsewhere). Hence, I don't think it is fair to compare them. A comparison should be systematic, so that the same surface composition and the same type of adsorption site are being compared against each other. If stretching causes restructuring, that should be commented on.

Response: The DFT calculation is carried out here to understand the change trend of HER performance of AC-Ir compare to A-Ir and C-Ir. According to the Arrhenius equation $k=Ae^{-E_a/RT}$, the change of activation energy will affect the rate of chemical reaction (k). Here $A = 2 \times 10^6$ [Angew. Chem. Int. Ed. 2018, 57, 7948-7956], $R = 8.314$ J mol⁻¹, $T = 300$ K. The influence of E_a change on the k change is shown in Figure R6. In this work, the ΔE_a is the adsorption energy difference, which is 0.03 eV. Clearly, the 0.03 eV changes for the adsorption energy of intermediates will lead to triple enhancement in the chemical reaction rate.

Figure R6. The ΔE_a -dependent k change.

We construct AC-Ir with different tensile strain based on the same initial surface and apply the tensile strain in the x direction with different ratio to study the strain effect. So, the number of C atoms per unit is same. To construct AC-Ir model, we replace 8 Ir atoms by C and the ratio of C:Ir is 8:64. The visual difference stems from the fact that some carbon atoms are obscured by the Ir atoms with larger atomic radius. Here, we mainly calculate the H adsorption on Ir and C top sites near the amorphous and crystalline interface as shown in Figure R2a. Due to the structural relaxation in the process of structural optimization, the optimized structures and atoms of AC-Ir with different tensile strain become disordered. Therefore, we calculate the adsorption energies of H^* on all designed Ir and C sites in different columns according to the optimized structures. It should be noted that some of sites have the same atomic environments, and only one of them are calculated, and some of sites have same energies. Overall, the calculated ΔG_{H^*} are shown in Figure R3.

Figure 4. I don't agree with the interpretation of Figure 4(a) in terms of the transfer of electrons from crystalline domains to amorphous domains in AC-Ir NSs. To me, it rather looks like the electrons are being transferred from Ir to a more electronegative C.

Response: The deformation charge density analysis of AC-Ir with different tensile strain is shown in Figure R7. We agree that the charge transfer essentially comes from Ir to more electronegative C atoms. We further calculate the Bader charge of each atom (Table R1). The amorphous domain is labelled by red circle in Figure R8. Here Ir1-Ir24, Ir31-Ir42, Ir49-Ir60 belong to crystalline domain, while Ir25-Ir30, Ir43-Ir48, Ir61-Ir64 and C1-C8 are belong to amorphous domain. The calculated Bader charge transfer from crystalline domains to amorphous domains (Δq) for AC-Ir with tensile strain from 0% to 20% are 1.52, 0.7, 0.7, 0.67, 0.65, 1.39 eV, respectively.

Figure R7. The deformation charge density analysis of AC-Ir with different tensile strain (isosurface is 0.12 bohr^{-3} , cyan represent charge depletion and yellow represent charge accumulation).

Figure R8. Schematic diagram of Bader charge transfer from crystalline domain to amorphous domain in AC-Ir NS.

Table1. Comparison of Bader charge of AC-Ir NSs with different degrees of lattice strain.

Element	AC-Ir TS-0%	AC-Ir TS-4%	AC-Ir TS-8%	AC-Ir TS-12%	AC-Ir TS-16%	AC-Ir TS-20%
Ir1	8.99306	9.05594	9.06568	9.07405	9.07692	9.08944
Ir2	8.99925	9.05663	9.06082	9.11244	9.13089	9.158
Ir3	8.79488	8.95801	8.87071	8.83168	8.82962	8.79968
Ir4	8.99263	9.05522	9.06603	9.07443	9.07627	9.08848
Ir5	8.99869	9.05674	9.0608	9.11206	9.13049	9.15872
Ir6	8.79441	8.95668	8.87007	8.83091	8.82635	8.79834
Ir7	9.0379	9.07332	9.06408	9.01772	9.01321	9.07714
Ir8	9.00502	9.00611	9.0254	9.0288	9.06363	9.13283
Ir9	8.93317	8.91082	8.92061	8.92513	8.91958	8.83992
Ir10	9.03841	9.07344	9.06297	9.01795	9.01346	9.07849
Ir11	9.00455	9.00608	9.02467	9.02815	9.06394	9.13218
Ir12	8.9334	8.91142	8.92038	8.92486	8.9199	8.84015
Ir13	8.86755	8.87526	8.8201	8.75404	8.78889	8.70984

Ir14	9.05556	9.08939	9.07131	9.08041	9.06523	8.96949
Ir15	9.07056	9.06102	9.04015	9.08858	9.07963	9.1061
Ir16	8.86918	8.87506	8.8203	8.75337	8.78975	8.7086
Ir17	9.05538	9.08939	9.07107	9.07959	9.06501	8.97023
Ir18	9.07083	9.06072	9.04064	9.0887	9.07963	9.10672
Ir19	8.9894	8.97046	8.94726	8.92743	8.89626	8.83662
Ir20	9.0875	9.07765	9.07203	9.0334	9.03516	9.0224
Ir21	9.03053	9.07176	9.01675	9.03581	9.04456	9.11726
Ir22	8.98789	8.9702	8.94837	8.9279	8.89711	8.83629
Ir23	9.08629	9.07796	9.07205	9.03354	9.03545	9.02306
Ir24	9.03006	9.07154	9.01721	9.03545	9.0458	9.11753
Ir25	8.64329	8.76164	8.83657	8.71921	8.72752	8.67864
Ir26	9.04411	8.80373	8.82182	8.82133	8.79951	8.82588
Ir27	8.85999	8.85338	8.76719	8.85144	8.83145	8.82095
Ir28	8.6437	8.7615	8.83715	8.71898	8.72638	8.68103
Ir29	9.04343	8.80395	8.82107	8.8208	8.80049	8.82639
Ir30	8.86345	8.85378	8.7665	8.85125	8.83169	8.82297
Ir31	8.92963	8.92436	8.96073	8.89664	8.93092	8.84115
Ir32	9.02454	9.00093	9.00937	8.99791	9.04343	9.06494
Ir33	9.0619	9.06517	9.08004	9.05095	9.01207	8.96049
Ir34	8.92896	8.92477	8.96092	8.89655	8.92962	8.84248
Ir35	9.02596	9.00142	9.00965	8.99777	9.04354	9.06505
Ir36	9.06163	9.06518	9.07991	9.05093	9.01211	8.95968
Ir37	8.83911	8.84818	8.85057	8.96844	8.97022	8.77949
Ir38	8.84246	9.01951	9.02029	8.97187	8.90945	8.90128
Ir39	8.89431	8.8727	9.03506	9.04203	9.05204	9.10762
Ir40	8.84015	8.84673	8.85068	8.96921	8.96983	8.7802
Ir41	8.84318	9.01895	9.02091	8.97217	8.90972	8.90131
Ir42	8.89557	8.87073	9.03475	9.04139	9.05255	9.1062
Ir43	8.73873	8.75899	8.77802	8.74138	8.73818	8.83191
Ir44	8.98656	8.86951	8.80681	8.88668	8.74039	8.93778
Ir45	8.84768	8.90535	8.88454	8.89429	8.87557	8.9711
Ir46	8.74526	8.76079	8.77786	8.74264	8.73816	8.83239
Ir47	8.98521	8.86928	8.80828	8.88781	8.73715	8.9357
Ir48	8.84827	8.90677	8.88346	8.89464	8.8759	8.97335
Ir49	9.04666	8.98998	9.00295	9.01668	8.99217	9.06157
Ir50	9.05936	9.08307	9.073	9.06007	9.05627	9.10098
Ir51	8.94098	8.9004	8.91662	8.92838	8.93099	8.89865
Ir52	9.04648	8.99018	9.00299	9.01593	8.99188	9.06199
Ir53	9.05905	9.08263	9.07317	9.06091	9.05513	9.09943
Ir54	8.94045	8.89994	8.91763	8.92926	8.93124	8.89801
Ir55	8.85638	8.86695	8.87766	8.99665	9.01637	9.05913
Ir56	8.89629	8.91619	8.86687	8.89596	8.88732	8.84904
Ir57	8.98381	8.9576	8.98249	8.92946	8.92942	8.8252

Ir58	8.85701	8.8666	8.8778	8.99671	9.015	9.05795
Ir59	8.89521	8.91669	8.86518	8.89463	8.88956	8.84697
Ir60	8.98408	8.95685	8.98212	8.92997	8.92925	8.82613
Ir61	8.82547	8.81124	8.90068	8.89091	9.1413	9.10271
Ir62	8.89068	8.82189	8.8989	8.89324	8.87676	8.87973
Ir63	8.82437	8.81068	8.90131	8.89095	9.14217	9.10089
Ir64	8.88364	8.82362	8.89869	8.89264	8.87418	8.88067
C1	4.49519	4.49331	4.4537	4.46639	4.46857	4.41011
C2	4.45866	4.43557	4.34856	4.32821	4.34374	4.4042
C3	4.46218	4.3782	4.48423	4.42368	4.40756	4.41756
C4	4.49662	4.49375	4.45363	4.46702	4.46855	4.40971
C5	4.45927	4.43318	4.35063	4.32963	4.34166	4.40256
C6	4.46242	4.37856	4.48538	4.42376	4.40838	4.41776
C7	4.50644	4.45779	4.3677	4.41811	4.37841	4.41204
C8	4.50613	4.45701	4.36645	4.41815	4.37952	4.41149
Δq	1.52	0.7	0.7	0.67	0.65	1.39

The English is in general good but some sentences suffer from wrong word choice and partially incorrect grammar. For example, I think that the expression “suitable Gibbs free energy” or “suitable H* adsorption” is wrong word choice. Probably, it should be “favorable” or “optimal” Gibbs free energy.

Here is another example. “Generalized gradient approximation (GGA) in the form of the Perdew–Burke–Ernzerhof (PBE) function with DFT-D3 scheme was used as the exchange-correlation interactions. Projector augmented wave (PAW) was employed to describe the interaction between valence electrons and ionic core.” This passage could be improved as follows: “The Perdew–Burke–Ernzerhof (PBE) form of the generalized gradient approximation (GGA) [ref] was used to describe the exchange-correlation functional. Dispersion effects have been accounted for according to Grimme's correction scheme (DFT-D3) [Grimme, S.; Antony, J.; Ehrlich, S. and Krieg, H., J. Chem. Phys., 2010, 132, 154104.] Projector augmented wave (PAW) method was employed to describe the interaction between valence electrons and ionic cores.”

“Comprehension on the effect of tensile strain” sounds a bit strange. I would change it to “Understanding the effect of tensile strain”.

I recommend proof-reading by a language expert who could improve the sentence structure. Because this is a top-tier journal, I also expect that the quality of written English is of sufficiently high standard.

Response: We thank the referee for carefully reviewing our manuscript. We have improved the language as the referee suggested, and we also ask our colleagues to help polish it.

REVIEWERS' COMMENTS

Reviewer #4 (Remarks to the Author):

The authors made a thoughtful and serious attempt to address the questions and critique points raised in my previous report. Importantly, in response to my review, they provided sufficient detail on the models, supported by appropriate figures. Although I don't necessarily agree with all of their conclusions, the work does represent a valuable contribution to the field and can be accepted without further review after the authors have addressed the following (minor) points.

1. Line 223: "As a proof, the standard hydrogen adsorption Gibbs free energy (ΔG_H^*), which is considered as the descriptor for HER activity, on the surface of AC-Ir NSs was also performed." I suggest not to use the word "proof" because it implies a rigorous proof, like in mathematics. Use the word "support" instead. I would suggest the following rephrasing: "To support our conclusions, we have calculated the standard hydrogen adsorption Gibbs free energy..."
2. Lines 226-227: It is stated that the H adsorption free energy of -0.06 eV (AC-Ir NSs with 4%) is "more favorable" than -0.23 eV (on Pt/C). However, the normal meaning of "more favorable" when it refers to adsorption refers to a state of lower relative energy and thus a more negative adsorption energy. I understand that the authors are referring to the work of Jaramillo et al. Science, 2007, 317, 100-102, which contains a volcano plot showing that the optimal H adsorption free energy is around 0 eV. This is an empirical observation and not something following directly from the laws of nature, therefore, it is important to explain the context and to cite their work. It will then be clear to the reader what "more favourable" refers to, but still better replace it with "more optimal for HER performance based on the conclusions of ref. [Jaramillo et al.]"
3. Caption of Figure 4. I would replace "Comprehension on the effect of tensile strain" by "Understanding the effect of tensile strain"
4. Lines 486-489: The formula for ΔG_H^* is now explained but I still have a question to the statement given after the formula. First, please, add units (eV) to the values of zero point vibrational energy, 0.17 and 0.135. Second, their difference is 0.035 eV which could be approximated as 0.04 eV, but is this value independent from the adsorption site of H and the type of the surface (C-Ir, A-Ir, AC-Ir)? If the value is the same in all cases, this should be explicitly stated. Currently, it is not specified, for which slab model and adsorption site these values are reported. Or are they taken from ref. 42? No calculation of vibrational frequencies is mentioned in the Theoretical Calculation Methods section, so I assumed the frequencies were not calculated or not for all geometries. If they were, please, mention this and comment on the ΔZPE for different adsorption sites and surface compositions.
5. Please, polish the English of the Theoretical Calculation Methods section. Some sentences are difficult to read. For example, I would suggest rewording the sentences in lines 463-466 as follows: "The crystalline Ir (0 -1 1) surface is represented by a periodic six-layer slab using a 3×2 surface unit cell containing 72 Ir atoms (12 atoms per layer). The Ir(0 -1 1) surface plane was chosen because it is perpendicular to the (2 0 0) and (1 1 1) planes, which compares well with the FFT patterns of crystalline structure in Figure 1."
6. When looking at top views of the Ir(0-11) surface (Supplementary figures 15, 18, 20), it is difficult to distinguish between the top layer and the second (subsurface layer). This could be improved by using a lighter color for subsurface Ir and C atoms or by using a different means to mark the top layer, e.g. contours of a certain color around atoms. In Fig. 25 this is done by using a smaller radius for subsurface atoms. This is also fine, but needs to be explained in the figure caption.
7. The following sentence in the caption of Suppl. Fig. 19 is unclear: "The unchanged structures with H* adsorption before and after optimization were provided into one graph." I suggest the following rephrasing: "Two structures, before and after optimization, are shown in those cases when the adsorption position of H changed during the optimization."

Point-by-point response to the referees' comments

We sincerely thank the referee for the careful review and valuable comments, which certainly help improve the manuscript. Our point-by-point responses are presented below and all the changes in the revised manuscript have been highlighted in red for your review.

Response to the Reviewer 4:

The authors made a thoughtful and serious attempt to address the questions and critique points raised in my previous report. Importantly, in response to my review, they provided sufficient detail on the models, supported by appropriate figures. Although I don't necessarily agree with all of their conclusions, the work does represent a valuable contribution to the field and can be accepted without further review after the authors have addressed the following (minor) points.

Response: We thank the reviewer very much for the positive comments and the recommendation for publishing.

1. Line 223: "As a proof, the standard hydrogen adsorption Gibbs free energy (ΔG_{H^*}), which is considered as the descriptor for HER activity, on the surface of AC-Ir NSs was also performed."; I suggest not to use the word "proof" because it implies a rigorous proof, like in mathematics. Use the word "support"; instead. I would suggest the following rephrasing: "To support our conclusions, we have calculated the standard hydrogen adsorption Gibbs free energy";

Response: We thank the referee for carefully reviewing our manuscript. Following the suggestion, we replaced '*As a proof*' with '*To support our conclusions*' in the revised manuscript.

2. Lines 226-227: It is stated that the H adsorption free energy of -0.06 eV (AC-Ir NSs with 4%) is "more favorable" than -0.23 eV (on Pt/C). However, the normal meaning of "more favorable" when it refers to adsorption refers to a state of lower relative energy and thus a more negative adsorption energy. I understand that the authors are referring to the work of Jaramillo et al. Science, 2007, 317, 100-102, which contains a volcano plot showing that the optimal H adsorption free energy is around 0 eV. This is an empirical observation and not something following directly from the laws of nature, therefore, it is important to explain the context and to cite their work. It will then be clear to the reader what "more favourable"; refers to, but still better replace it with "more optimal for HER performance based on the conclusions of ref. [Jaramillo et al.]";

Response: We thank the referee for carefully reviewing our manuscript. Following the suggestion, we replaced '*Figure 4c reveals the H* adsorption on surface of AC-Ir NSs with 4% tensile strain (-0.06 eV) is more favorable than those on A-Ir NSs (0.22 eV), C-Ir NSs (-0.37 eV) and Pt/C catalysts (-0.23 eV)*' with '*Figure 4c reveals the H* adsorption on surface of AC-Ir NSs with 4% tensile strain (-0.06 eV) is more optimal than those on A-Ir NSs (0.22 eV), C-Ir NSs (-0.37 eV) and Pt/C catalysts (-0.23 eV), based on the conclusions of the HER volcano plot*' in the revised manuscript.

3. Caption of Figure 4. I would replace “Comprehension on the effect of tensile strain” by “Understanding the effect of tensile strain”.

Response: We thank the reviewer for the valuable suggestions. Following the suggestion, we replaced ‘*Comprehension on the effect of tensile strain*’ with ‘*Understanding the effect of tensile strain*’ in the revised the caption of Figure 4 in the revised manuscript.

4. Lines 486-489: The formula for ΔG_{H^*} is now explained but I still have a question to the statement given after the formula. First, please, add units (eV) to the values of zero point vibrational energy, 0.17 and 0.135. Second, their difference is 0.035 eV which could be approximated as 0.04 eV, but is this value independent from the adsorption site of H and the type of the surface (C-Ir, A-Ir, AC-Ir)? If the value is the same in all cases, this should be explicitly stated. Currently, it is not specified, for which slab model and adsorption site these values are reported. Or are they taken from ref. 42? No calculation of vibrational frequencies is mentioned in the Theoretical Calculation Methods section, so I assumed the frequencies were not calculated or not for all geometries. If they were, please, mention this and comment on the ΔZPE for different adsorption sites and surface compositions.

Response: Thanks for your valuable suggestion. Here the value of $\Delta ZPE - T\Delta S_H$ is taken from ref. 42 (*J. Electrochem. Soc.* 2005, 152, J23-J26). The contribution from the vibrational frequencies is negligibly small with harmonic approximation. So the contribution from slabs are neglected here. This is an approximation formula, because the difference between the adsorption energy (at 0 K in vacuum) and ΔG is given by a constant term, same for all adsorption sites. Such approximation has also been carried out for the ΔG_{H^*} correction on the surfaces of transition metal and metal cluster (*Nat. Commun.* 2020, 11, 5462. *Nat. Commun.* 2022, 13, 2430. *Chem. Eng. J.* 2022, 446, 137297.).

5. Please, polish the English of the Theoretical Calculation Methods section. Some sentences are difficult to read. For example, I would suggest rewording the sentences in lines 463-466 as follows: “The crystalline Ir (0 -1 1) surface is represented by a periodic six-layer slab using a 3x2 surface unit cell containing 72 Ir atoms (12 atoms per layer). The Ir(0 -1 1) surface plane was chosen because it is perpendicular to the (2 0 0) and (1 1 1) planes, which compares well with the FFT patterns of crystalline structure in Figure 1.”

Response: We thank the referee for the constructive comments. Following the suggestion, we have revised the description about Theoretical Calculation Methods section in the revised manuscript and we also ask our colleagues to help polish it again.

6. When looking at top views of the Ir(0-11) surface (Supplementary figures 15, 18, 20), it is difficult to distinguish between the top layer and the second (subsurface layer). This could be improved by using a lighter color for subsurface Ir and C atoms or by using a different means to mark the top layer, e.g. contours of a certain color around

atoms. In Fig. 25 this is done by using a smaller radius for subsurface atoms. This is also fine, but needs to be explained in the figure caption.

Response: Thanks for your valuable suggestion. Following the suggestion, we use orange, grey, light orange and light grey spheres represent surface iridium, surface carbon, subsurface iridium and subsurface carbon atoms in the slab models, respectively. Furthermore, we have revised the related figures (Supplementary figures 15, 18, 20 and 25) and added more description of the models in the revised supporting information.

7. The following sentence in the caption of Suppl. Fig. 19 is unclear: “The unchanged structures with H* adsorption before and after optimization were provided into one graph.”; I suggest the following rephrasing: “Two structures, before and after optimization, are shown in those cases when the adsorption position of H changed during the optimization.”

Response: We thank the referee for carefully reviewing our manuscript. Following the suggestion, we replaced ‘*The unchanged structures with H* adsorption before and after optimization were provided into one graph.*’ with ‘*Two structures, before and after optimization, are shown in those cases when the adsorption position of H* changed during the optimization.*’ in the revised caption of Supplementary fig. 19 and 22.